EMBO
Molecular Medicine

# Glufosinate constrains synchronous and metachronous metastasis by promoting anti-tumor macrophages

Alessio Menga[1,2,3,4] (iD), Marina Serra[1,2], Simona Todisco[5], Carla Riera-Domingo[1,2], Ummi Ammarah[3], Manuel Ehling[1,2], Erika M Palmieri[6], Maria A Di Noia[4], Rosanna Gissi[4], Maria Favia[4], Ciro L Pierri[4], Paolo E Porporato[3], Alessandra Castegna[4,7,*] (iD) & Massimiliano Mazzone[1,2,3,**] (iD)

## Abstract

Glutamine synthetase (GS) generates glutamine from glutamate and controls the release of inflammatory mediators. In macrophages, GS activity, driven by IL10, associates to the acquisition of M2-like functions. Conditional deletion of GS in macrophages inhibits metastasis by boosting the formation of anti-tumor, M1-like, tumor-associated macrophages (TAMs). From this basis, we evaluated the pharmacological potential of GS inhibitors in targeting metastasis, identifying glufosinate as a specific human GS inhibitor. Glufosinate was tested in both cultured macrophages and on mice bearing metastatic lung, skin and breast cancer. We found that glufosinate rewires macrophages toward an M1-like phenotype both at the primary tumor and metastatic site, countering immunosuppression and promoting vessel sprouting. This was also accompanied to a reduction in cancer cell intravasation and extravasation, leading to synchronous and metachronous metastasis growth inhibition, but no effects on primary tumor growth. Glufosinate treatment was well-tolerated, without liver and brain toxicity, nor hematopoietic defects. These results identify GS as a druggable enzyme to rewire macrophage functions and highlight the potential of targeting metabolic checkpoints in macrophages to treat cancer metastasis.

**Keywords** glufosinate; glutamine synthetase; immunometabolism; macrophages; metastasis

**Subject Categories** Cancer; Metabolism; Pharmacology & Drug Discovery

## Introduction

Tumor-associated macrophages (TAMs) constitute the main stromal compartment in many tumors. TAMs support different crucial functions for cancer progression by shaping adaptive immune responses and by contributing to vessel and matrix remodeling, cancer cell proliferation, survival, and metastasis (Noy & Pollard, 2014; Singh *et al*, 2017). At the metastatic site, macrophages (that are metastasis-associated macrophages, MAMs) can also favor the preparation of the pre-metastatic niche and cancer cell extravasation (Gil-Bernabé *et al*, 2012; Sharma *et al*, 2015; Lin *et al*, 2019). For these reasons, targeting macrophage functions is now considered a promising strategy to harness primary and metastatic tumor growth. More evidence on TAMs suggests but does not always prove that these diverse functions are driven and sustained by distinct metabolic states (Mazzone *et al*, 2018; Flerin *et al*, 2019; Castegna *et al*, 2020). The classical pro-inflammatory and antitumoral macrophage phenotype can be resembled by *in vitro* stimulation with lipopolysaccharide (LPS)/interferon-γ (IFN-γ), whereas the alternative, pro-tumor function is achieved by IL4 and IL10. These cytokines are responsible for metabolic shifts that underline defined functional states in macrophages (Biswas & Mantovani, 2012; Mills & O'Neill, 2016). More information is awaited regarding the role of specific metabolic features affecting TAM behavior *in vivo*, the strategies to rewire their protumoral phenotype toward their original protective, anti-tumor function, and their impact on disease outcome (Wenes *et al*, 2016; Palmieri *et al*, 2017b). In the effort to dissect which pathways underline the phenotypic switch in TAMs, we have discovered that targeting glutamine synthetase (GS; a.k.a. glutamate ammonia ligase, GLUL, EC 6.3.1.2) in macrophages represents an effective way to reprogram *in vitro* IL10-stimulated macrophages and TAMs toward a desirable "M1-like function". GS-inhibited IL10 macrophages display

1 Laboratory of Tumor Inflammation and Angiogenesis, Center for Cancer Biology (CCB), VIB, Leuven, Belgium
2 Laboratory of Tumor Inflammation and Angiogenesis, Department of Oncology, KU Leuven, Leuven, Belgium
3 Department of Molecular Biotechnology and Health Science, Molecular Biotechnology Centre, University of Torino, Torino, Italy
4 Department of Biosciences, Biotechnologies and Biopharmaceutics, University of Bari, Bari, Italy
5 Department of Sciences, University of Basilicata, Potenza, Italy
6 Cancer & Inflammation Program, National Cancer Institute, Frederick, MD, USA
7 IBIOM-CNR, Institute of Biomembranes, Bioenergetics and Molecular Biotechnologies, National Research Council, Bari, Italy
*Corresponding author. Tel: +39 080 5442322; E-mail: alessandra.castegna@uniba.it
**Corresponding author. Tel: +32 16 37 3213; E-mail: massimiliano.mazzone@vib-kuleuven.be

typical M1-like features both metabolically and functionally, with an increased glycolytic flux partly linked to HIF1α stabilization, enhanced ability to impair cancer invasion, reduced capacity to induce angiogenesis, and enhanced propensity for T-cell activation and recruitment. This ultimately results in metastasis inhibition in tumor-bearing mice (Palmieri *et al*, 2017b). These encouraging results have prompted us to pursue inhibition of GS in macrophages as an immunometabolic strategy to reduce metastasis in cancer.

Methionine sulfoximine (MSO) is the classical GS inhibitor through irreversible binding to the glutamate site, but many studies have been performed on different classes of molecules able to inhibit GS, with particular attention to plant GS, in which inhibition of glutamine synthesis has been proved particularly effective in killing weeds (Occhipinti *et al*, 2010). Glufosinate, or phosphinothricin [CAS Number: 51276-47-2], is a non-proteinogenic amino acid that inhibits GS in bacteria (Bayer *et al*, 1972), algae (Hall *et al*, 1984), and higher plants (Leason *et al*, 1982). Studies in rats have shown that pharmacological inhibition of GS by glufosinate ammonium is lethal at very high doses ($LD_{50}$ 1,500–2,000 mg/kg); however, our understanding on the systemic toxicity of this molecule (e.g., $TD_{50}$ studies) remains sparse and not conclusive (Cox, 1996). Glufosinate is a glutamate analogue and competes with glutamate for the binding in the active site (Logusch *et al*, 1989). The effect of glufosinate on human GS has never been proven, but the specificity of its inhibitory action in plant GS (Occhipinti *et al*, 2010) prompted us to test its effect on the human recombinant GS as well as in macrophages both *in vitro* and *in vivo*. Our results point to a significant inhibitory capacity of glufosinate toward human GS. Furthermore, at micromolar concentrations glufosinate treatment induces a shift from a M2-like to an M1-like phenotype in murine and human macrophages similarly to the effect displayed by MSO. This rewiring is associated with a decreased immunosuppression, angiogenesis, and metastatic burden in murine models of lung, breast cancer, and melanoma treated with glufosinate, similarly to what observed in the macrophage-specific GS knockout mice (Palmieri *et al*, 2017b), without any sign of specific liver or brain toxicity nor hematopoietic abnormalities at all tested doses.

In the clinic, cancer death is mostly due to the appearance of diffused metastatic lesions that can be treated together with the primary tumor when the latter is unresectable or already in advanced stages (synchronous metastasis). Alternatively, metastasis treatment can start few or several years after primary tumor resection following the relapse of the disease far from the original site (metachronous metastasis) (Mekenkamp *et al*, 2010). The data hereby presented show that the antimetastatic effect of glufosinate is evident not only in a synchronous setting but also in a metachronous setting, increasing the clinical potential of our findings. This study provides a proof-of-concept of the role of *in vivo* pharmacological GS inhibition as immunometabolic strategy to reduce metastasis and highlights the significance of *in vivo* targeting of macrophagic metabolic checkpoints as a promising alternative to tackle tumor progression and metastasis.

# Results

## Glufosinate inhibits human recombinant GS

The enzyme activity of the human recombinant GS (hGS), obtained by its expression in *E. coli* (Fig 1A), was evaluated in the presence of saturating concentrations of substrates for almost 90 min at 25°C and was found to be linear at least for 10 min at 25°C (Fig 1B). The Michaelis–Menten (half-saturation) constant ($K_m$) of the human recombinant GS was determined by measuring the initial rate by varying L-glutamate or ATP concentrations in presence of a fixed saturating concentration of the other two substrates. By Lineweaver–Burk plots, the $K_m$ and $V_{max}$ values were 1.42 ± 0.17 mM and 0.82 ± 0.05 µmol/min/mg protein for L-glutamate, and 2.56 ± 0.15 mM and 0.91 ± 0.03 µmol/min/mg protein for ATP (Fig 1C and D).

The effect of glufosinate on GS activity was investigated and was compared to that displayed by MSO on the human recombinant protein. Glufosinate and MSO inhibited human recombinant GS in a concentration-dependent manner (Fig 1E). Interestingly, glufosinate inhibition was much more effective compared to that of MSO as

---

**Figure 1. Glufosinate specifically inhibits human recombinant GS.**

A   Expression in *E. coli* and purification of human GS (hGS). Protein was separated by SDS–PAGE and stained with Coomassie Blue dye. M: marker Precision Plus Protein Dual Color Standard (Bio-Rad). From right Lanes 1–4: *E. coli* Bl21(DE3) cells containing the expression vector without (lanes 1 and 2) and with (lanes 3 and 4) the coding sequence of hGS. Samples were taken immediately before (lanes 1 and 3) and 3.5 h later (lanes 2 and 4) the induction of expression with isopropyl-β-D-1-thiogalattopiranoside (IPTG) 0.7 mM. The same number of bacteria was analyzed in each sample. Lane 5: isolated and purified inclusion bodies (4 µg). Adjacent boxed lane: Western blotting analysis of GS.

B–E   Kinetic study of the reaction catalyzed by recombinant hGS. The reaction, started by adding ATP, was linear for at least 10 min at 25°C (B). Lineweaver–Burk plot reporting the hGS activity at the indicated concentrations of glutamate in the absence (●), or in the presence of glufosinate (C) or MSO (D). Symbols (○): 2 mM MSO or 0.025 glufosinate; (■) 3 mM of MSO or 0.050 mM of glufosinate; (□) 4 mM of MSO or 0.065 mM of glufosinate; (◆) 5 mM of MSO or 0.075 mM of glufosinate. The insets represent the secondary plot of the slopes of Lineweaver–Burk plot obtained at the indicated concentrations of glufosinate (C) or MSO (D) used for determining the inhibitor constant Ki. GS inhibition (%) in presence of increasing concentrations of glufosinate (○) or MSO (●) results from the average of at least three independent experiments (E). The control value for uninhibited hGS activity is 0.51 ± 0.08 µmol/(min × mg protein).

F–K   Comparative analysis of GS structures from several organisms. Lateral (F) and top view (G) of the entire GS decameric (a dimer of pentamers) structure are reported in pink cartoon representation. (H) Cartoon representation showing the top view of the monomer–monomer interface hosting cofactors, substrates, or inhibitors (Dataset EV1) participating to or inhibiting the conversion of glutamate to glutamine. In particular, ADP (cyan), phosphate ions (brown), MSO phosphate (P3S, yellow), phosphoaminophosphonic acid-adenylate ester (ANP, white), glutamate (magenta), citrate (black), and the imidazopyridine inhibitor ((4-(6-bromo-3-(butylamino)imidazo(1,2-a)pyridin-2-yl)phenoxy) acetic acid, green) are reported in sticks representation, whereas $Mn^{2+}$ ions are reported in pale blue spheres and $Mg^{2+}$ in pale green spheres (involved in the coordination of the imidazopyridine inhibitor). (I) Superimposition of all the sampled 17 crystallized structures (Dataset EV1). (J) 2D representation of glutamate, P3S, glufosinate, and glyphosate. (K) Zoomed view of the crystallized P3S binding region and the docked glufosinate and glutamate binding regions within the superimposed human (white cartoon) and *Z. mays* (pink cartoon) GS structures.

Source data are available online for this figure.

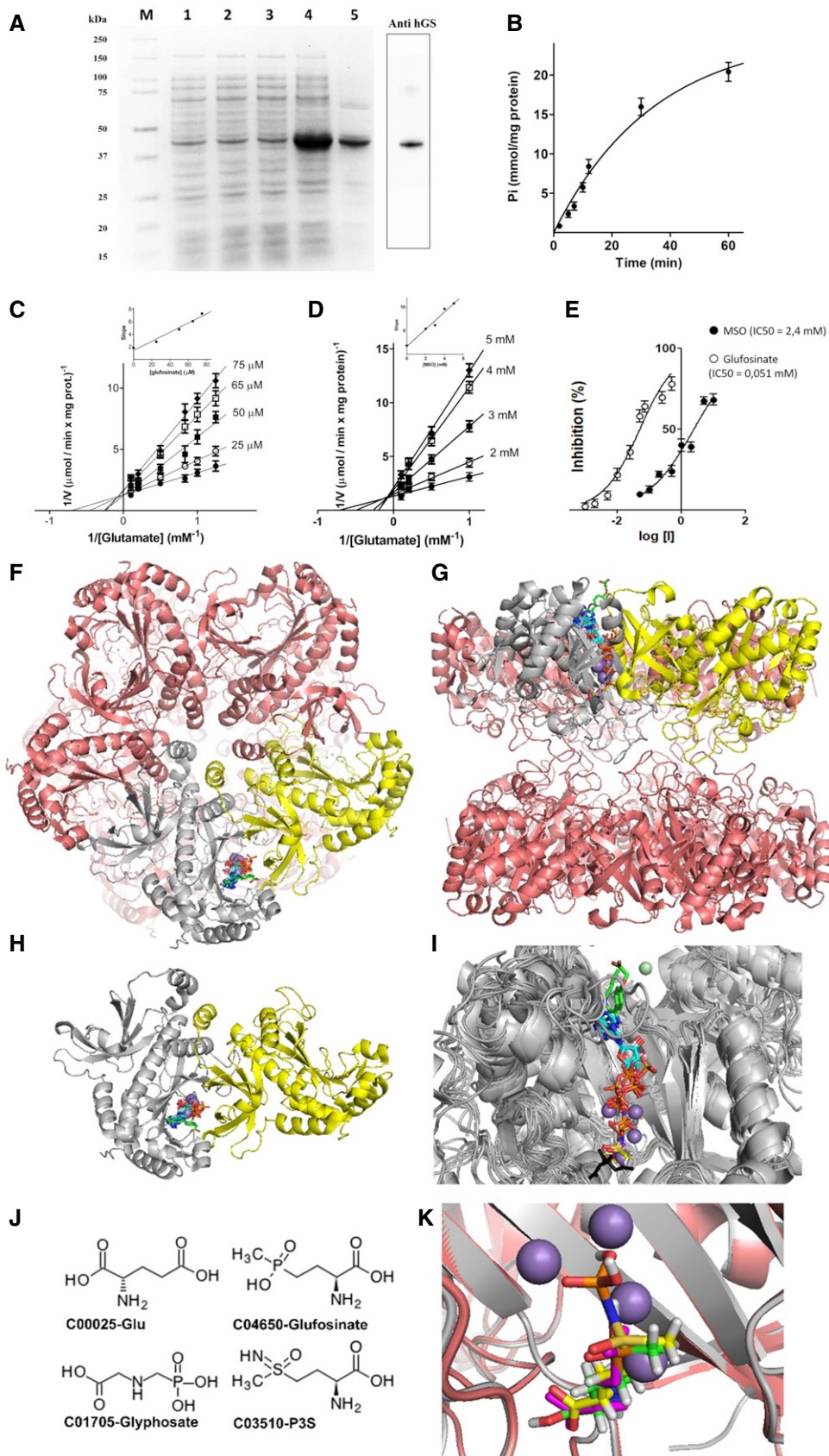

Figure 1.

showed by the IC$_{50}$ values that are of 0.051 and 2.4 mM, respectively (Fig 1E).

To shed light into the inhibition mechanism, double-reciprocal plots were obtained from the reciprocal of initial rates versus the reciprocal of L-glutamate with or without different concentrations of MSO or glufosinate. The initial binding of glufosinate (Fig 1C) or MSO (Fig 1D) to human GS was competitive versus glutamate. The Ki value for MSO, estimated by secondary plot, was 0.88 mM in good agreement with the value previously obtained (Jeitner & Cooper, 2014) (inset of Fig 1D). Furthermore, the Ki value of glufosinate was estimated about 0.0195 mM (inset of Fig 1C) confirming a greater selectivity of this inhibitor when compared to that of MSO.

To rule out an additional off-target role of glufosinate, a sequence database screening was performed by using alternatively *H. sapiens* or *Z. mays* GS sequences, searching for highly similar paralogous sequences in mammalia and plants (Dataset EV1). This search revealed no significant similarity between the query GS sequences and other *mammalia* or *plant paralogous* sequences, different from GS, nor identified structurally related proteins, beyond the available GS proteins, in the PDB until today (Dataset EV2).

For gaining new insights about glufosinate competitive binding mechanism to GS binding region, in presence of glutamate, the available crystallized structures were manually inspected by using 3D visualizer. Thus, it was observed that several ligands participating in the reaction catalyzed by GS were crystallized in complex at the monomer–monomer interface with GS (Fig 1F–K), namely ADP, phosphate ion, L-methionine-S-sulfoximine phosphate (P3S), phosphoaminophosphonic acid-adenylate ester (ANP), glutamate, citrate, and the imidazopyridine inhibitor ((4-(6-bromo-3-(butylamino)imidazo(1,2-a)pyridin-2-yl)phenoxy) acetic acid) (Dataset EV2). Given the structural similarity of glufosinate with glutamate ligand and P3S, after superimposition

of the three molecules in the GS catalytic site, it is observed that glufosinate might bind most of residues involved in the binding of glutamate ligand (in 4hpp.pdb) and P3S (i.e., in 2qc8.pdb) (Fig 1F–K and Dataset EV3).

### Glufosinate skews macrophages away from an M2-like phenotype and promotes an M1-like phenotype *in vitro*

Based on the inhibitory capacity of glufosinate on the human recombinant protein, together with our previous findings on the role of GS in macrophages (Palmieri *et al*, 2017b), we asked whether pharmacological GS targeting affects polarization of primary human macrophages. To this end, macrophages derived from human monocytes were skewed toward an M2-like phenotype with IL10 in presence or absence of glufosinate and the expression levels of M1 and M2 markers were measured. In IL10-stimulated macrophages (IL10-macrophages for simplicity), M1-like genes, such as *CD80*, *CXCL9,* and *CXCL10* (Fig 2A–C), were upregulated by glufosinate in a concentration-dependent fashion, except for *TNFA*, which strongly increased only at 20 μM concentration (Fig 2D). Concomitantly, up-regulation of M2-specific markers upon IL10 stimulation, such as *MSR1* (*CD204*) and *MRC1* (*CD206*), *CCL17* and *CCL18,* was reduced (Fig 2E–H). These data indicate that glufosinate effectively blocks GS activity in macrophages and this prevents the expression of M2 markers while promoting the acquisition of M1 features, in a more effective fashion compared to MSO (Palmieri *et al*, 2017b). From a metabolic point of view, glufosinate/IL10 macrophages displayed increased glutamate and succinate levels, with a decrease in glutamine compared to IL10 macrophages (Fig EV1A), phenocopying the metabolic reprogramming displayed by MSO-treated IL10 macrophages (Palmieri *et al*, 2017b).

---

**Figure 2. Glufosinate promotes a rewiring of IL10 macrophages toward a M1-like phenotype through HIF1α stabilization and abolishes immunosuppressive effect of hypoxia.**

A–D Evaluation of M1 markers in macrophages by real-time PCR. Fold change of *TNFA, CD80, CXCL9,* and *CXCL10* mRNA in IL10, MSO- and glufosinate (10 and 20 μM)-stimulated IL10 macrophages (*n* = 3).

E–H Evaluation of M2 markers in macrophages by real-time PCR. Fold change of *MRC1, MSR1, CCL17,* and *CCL18* mRNA in IL10, MSO-, and glufosinate (10 and 20 μM)-stimulated IL10 macrophages (*n* = 3).

I–M Evaluation of M1 markers in macrophages following HIF1α inhibition by real-time PCR. Fold change of *TNFA, CXCL10, CD86, CD80,* and *CXCL9* mRNA in IL10 alone or glufosinate (10 and 20 μM)- and acriflavine/glufosinate (10 and 20 μM)-IL10 macrophages (*n* = 3).

N–P Evaluation of M2 markers in macrophages following HIF1α inhibition by real-time PCR. Fold change of *MRC1, MSR1,* and *CCL18* mRNA in IL10, glufosinate (10 and 20 μM)-treated, and acriflavine/glufosinate (10 and 20 μM)-treated IL10 macrophages (*n* = 3).

Q Quantification of cancer cell motility through a matrigel-coated membrane in presence of IL10, MSO/IL10, and glufosinate (10 and 20 μM)-IL10-treated macrophages after 24 h of incubation (*n* = 6).

R Evaluation of the capillary network formation in presence of macrophages pretreated for 24 h with IL10 or MSO/IL10, and glufosinate (10 and 20 μM)/IL10 after 4 h of incubation with HUVEC cells (*n* = 6).

S CD8$^+$ T-cell suppression by macrophages treated with IL10 or MSO/IL10, and glufosinate (10 and 20 μM)/IL10 for 24 h (*n* = 4). Proliferation was evaluated by reading radioactivity as cpm (counts per minute), after incubation with 1 μCi/well tritiated thymidine. The proliferation of T cells cultured without macrophages was used as control.

T CD8$^+$ T-cell recruitment in a transwell system by macrophages treated with IL10 or MSO/IL10, and glufosinate (10 and 20 μM)/IL10 for 24 h versus macrophages treated with LPS/IFNγ; the migration of T cells cultured without macrophages (Mφ-) in presence of CXCL10 was used as positive control (*n* = 4).

U Representative image of Western blotting analysis of HIF1α, REDD1, 4E-BP1, S6, and P70S6K (in their phosphorylated and unphosphorylated form) to test mTOR activation in normoxic (NRX) and hypoxic (HYP) IL10 macrophages treated with glufosinate (20 μM), rapamycin (20 nM) and a combination of both (*n* = 3).

V CD8$^+$ T-cell suppression by normoxic and hypoxic IL10 macrophages treated with glufosinate (20 μM), rapamycin, and a combination of both for 24 h (*n* = 4). Proliferation was evaluated by reading radioactivity as cpm (counts per minute), after incubation with 1 μCi/well tritiated thymidine. The proliferation of T cells cultured without macrophages was used as control.

Data information: Data are reported as means ± SEM. *$P < 0.05$, **$P < 0.01$, ***$P < 0.001$, ****$P < 0.0001$. Exact $P$ values and statistical tests are reported for each experiment in Appendix Table S2.
Source data are available online for this figure.

---

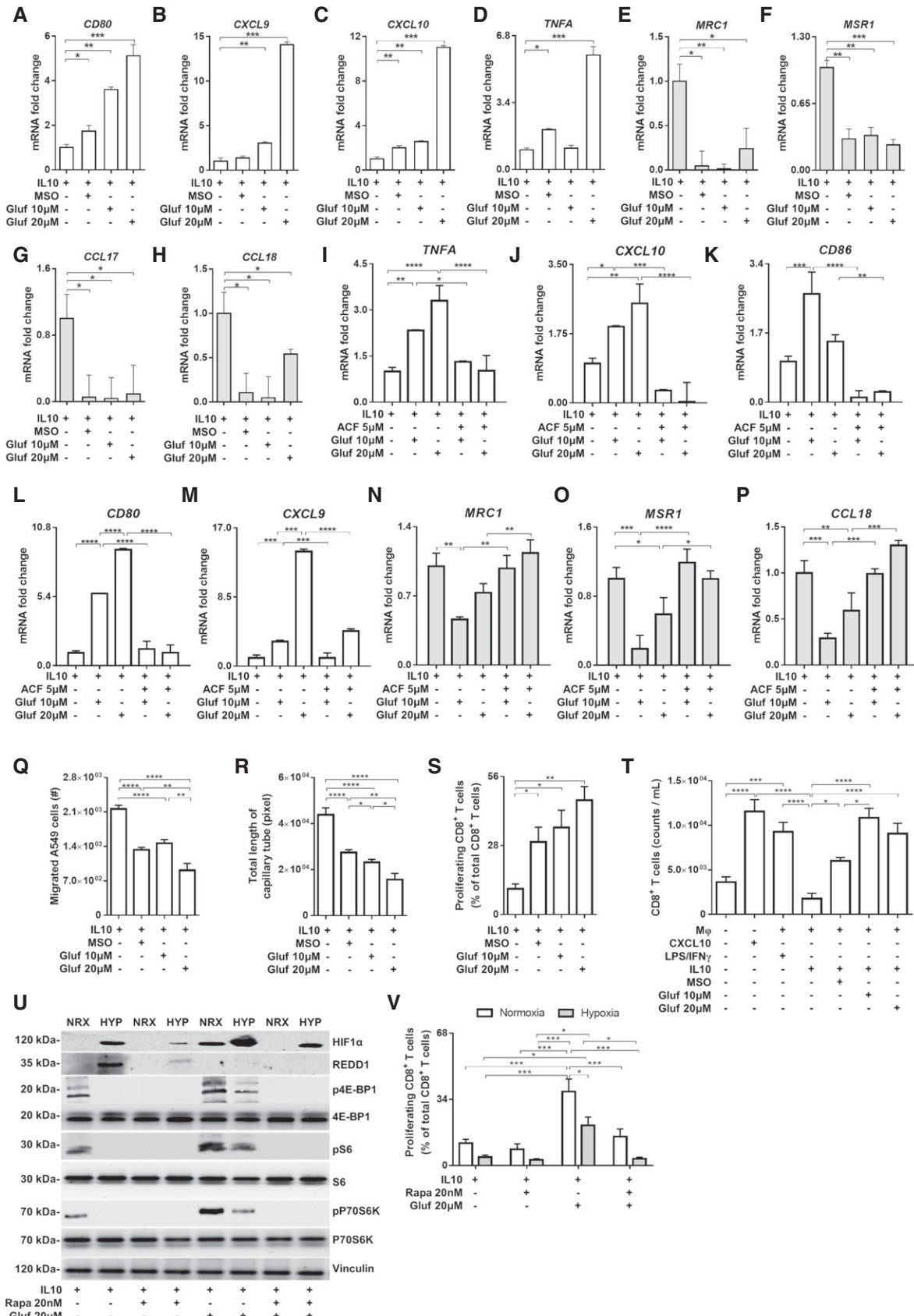

Figure 2.

### Glufosinate-treated macrophages display HIF1α activation and relevant anti-tumor functions *in vitro*

Since glufosinate promotes succinate accumulation and skews IL10 macrophages toward an M1-like phenotype, which is characterized by succinate accumulation (Fig EV1A), we tested whether HIF1α, known to be stabilized by succinate (Tannahill *et al*, 2013), is upstream to the expression of a pro-inflammatory M1 phenotype (Takeda *et al*, 2010). Treatment with the HIF1α inhibitor acriflavine prevented the M2 to M1 phenotypical rewiring in glufosinate-treated

IL10 macrophages, since the expression of typical markers of classically activated macrophages (such as *TNFA, CXCL10, CD86, CD80,* and *CXCL9*) was significantly decreased (Fig 2I–M). Concomitantly, acriflavine reduced the expression of markers expressed in IL10 macrophages such as *MRC1, MSR1,* and *CCL18* (Fig 2N–P). This trend was confirmed at the protein level for TNFα and CCL18 (Fig EV1B). Since it is known that M2-like macrophages support cancer cell migration (Joyce & Pollard, 2009), we evaluated the extent of cancer motility through 8 μm-pores in presence of macrophages treated for 24 h with IL10 versus glufosinate/or MSO/IL10

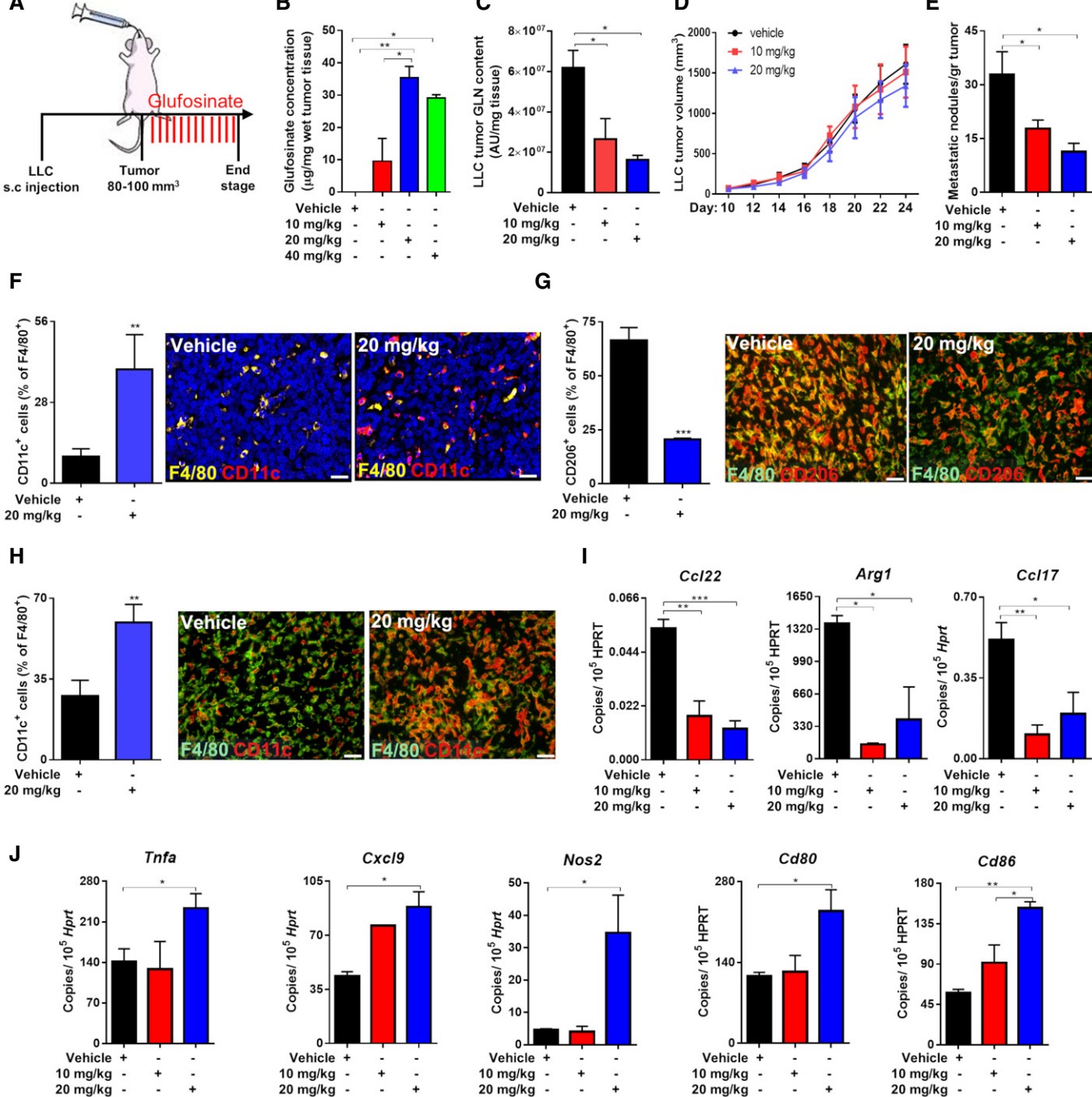

**Figure 3.**

◄

**Figure 3. Glufosinate treatment induces an M1-like phenotype and inhibits metastasis.**

A Experimental plan. Mice were subcutaneously injected with LLC cells. When the tumor reached 80–100 mm$^3$ size, mice were treated by gavage with glufosinate (10, 20, and 40 mg/kg) for 14 days.

B, C Quantification of glufosinate (B) and glutamine (GLN) (C) levels in wet tumor tissues of vehicle and glufosinate-treated mice ($n = 8$), by liquid chromatography–mass spectrometry (LC-MS).

D, E Subcutaneous LLC tumor growth over the time (D) and number of lung metastatic nodules in vehicle and glufosinate (10 and 20 mg/kg)-treated mice (E) (pool of 2 independent experiments; 10 mice per condition in total).

F–H Quantification and representative images of F4/80$^+$ CD11c$^+$ cells in the lung metastasis (F), F4/80$^+$ CD206$^+$ (G) and F4/80 CD11c$^+$ (H) cells infiltration in tumors of vehicle and glufosinate-treated mice ($n = 6$).

I, J RT–PCR quantification of M2 (*Ccl22, Arg1, and Ccl17*) (I) and M1 (*Tnfa, Cxcl9, Nos2, Cd86, and Cd80*) (J) markers in vehicle and glufosinate-treated mice ($n = 4$).

Data information: In each histological quantification, $n$ represents the number of animals. Six images per tumor were analyzed. Scale bars: 20 μm (F), 50 μm (G, H). Data are reported as means ± SEM. *$P < 0.05$, **$P < 0.01$, ***$P < 0.001$, ****$P < 0.0001$. Exact $P$ values and statistical tests are reported for each experiment in Appendix Table S2.

Source data are available online for this figure.

and then washed out. In line with the finding that glufosinate hinders M2-like features, A549 human lung cancer cells in the presence of glufosinate/IL10 macrophages were significantly less motile in a concentration-dependent fashion in comparison to cells exposed to IL10-stimulated control macrophages (Figs 2Q and EV1C).

Finally, we assessed the capillary network formation in response to M2-like macrophages with or without glufosinate. Compared to IL10 macrophages, macrophages pre-stimulated with IL10 and glufosinate were less effective in promoting capillary formation, suggesting that glufosinate hinders in a concentration-dependent fashion the ability of IL10 macrophages to sustain vessel sprouting (Figs 2R and EV1D).

We then tested the role of glufosinate in modulating CD8$^+$ T-cell proliferation and migration. To this end, we treated IL10 macrophages with vehicle, MSO, or glufosinate, and after washing the treatment out, we reseeded each condition with autologous CD8$^+$ T cells. Upon 24 h stimulation, IL10 macrophages displayed the strongest effect in suppressing the proliferation of cocultured CD8$^+$ T cells. Glufosinate blunted T-cell suppression imposed by IL10 macrophages in a concentration-dependent fashion and more effectively than MSO, as shown by CD8$^+$ T-cell proliferation partial rescue when glufosinate (or MSO)-stimulated IL10 macrophages were cocultured with pre-activated T cells (Fig 2S). To quantify T-cell adhesion and chemotaxis, CD8$^+$ lymphocytes were cultured alone, in the presence of CXCL10, LPS/IFNγ, IL10, MSO/IL10, and glufosinate/IL10 macrophages. T-cell chemotaxis through a 5 μm pore membrane was promoted by CXCL10 or by M1 macrophages, but not by IL10-macrophages and the condition without macrophages (Fig 2T). However, when cultured with glufosinate-treated IL10 macrophages, T cells were recruited and were able to migrate to a similar extent as those cultured with LPS/IFNγ macrophages (Fig 2T).

Although stabilization of HIF1α in normoxic macrophages is known to sustain a pro-inflammatory, M1-like phenotype (Tannahill *et al*, 2013), in hypoxia HIF1α is exploited by TAMs to suppress T cells via iNOS induction (Doedens *et al*, 2010). In oxygen and nutrient-replete conditions, TORC1 is known to sustain a pro-inflammatory phenotype (Byles *et al*, 2013; Zhu *et al*, 2014; Moon *et al*, 2015). However, under hypoxia, mTOR signaling is turned off and this occurs partly via the transcriptional induction of REDD1 (Wenes *et al*, 2016). We hypothesized that GS inhibition in hypoxic macrophages leads to mTOR re-activation via REDD1 blockade (Mazzone *et al*, 2018), ultimately promoting the acquisition of normoxic M1-like features linked to HIF1α stabilization (and TORC1 signaling) in

macrophages. Indeed, REDD1 protein levels were higher in IL10 hypoxic versus normoxic macrophages, leading to mTOR inhibition (Fig 2U). However, glufosinate treatment reverted this effect, as REDD1 protein levels decreased and mTOR got activated despite hypoxia (Fig 2U). This eent promoted the acquisition of classical normoxic M1-like features of IL10-macrophages that, even in hypoxic conditions, still sustain T-cell activation to a significantly higher extent compared to IL10 macrophages (Fig 2V).

Hence, glufosinate prevents the protumoral effects of IL10-stimulated macrophages by improving CD8$^+$ T-cell proliferation and migration, and by inhibiting capillary formation and cancer cell invasion.

## Glufosinate treatment inhibits metastasis in different cancer types

In order to translate our findings in mouse models of cancer, we first assessed the effects of glufosinate in murine macrophages isolated from BMDMs. Similar to the results on human IL10 macrophages, glufosinate/IL10 murine macrophages displayed upregulated M1 markers, such as *Tnfa, Cxcl10,* and *Nos2* (Fig EV2A). Glufosinate blocked the up-regulation of M2-specific markers following IL10 stimulation, such *Ccl22* and *Arg1,* the latter displaying a more significant reduction at increasing glufosinate concentrations (Fig EV2B). Coculture experiments with Lewis lung carcinoma (LLC) cells showed that glufosinate impaired the effect of murine IL10 macrophages on LLC proliferation and migration (Fig EV2C and D), although the concentrations of glufosinate used did not exert any differential effect. Glufosinate treatment directly on LLC cells did not show an effect on proliferation and migration both in normal medium and under glutamine deprivation (Fig EV2E). These results were obtained also with 4T1 breast cancer (Fig EV2F) and YUMM1.7 melanoma cells (Fig EV2G), used in further *in vivo* experiments.

Therefore, we tested the effect of glufosinate on mice bearing different cancer types. As a lung cancer model, LLC cells were implanted subcutaneously in C57Bl/6J mice and followed for tumor growth (Fig 3A). Glufosinate levels, measured within the tumor mass, were increased in a concentration-dependent fashion up to 20 mg/kg, above which saturation occurred (Fig 3B). Given the saturation in intratumoral glufosinate levels reached at 20 mg/kg, we decided to test the *in vivo* efficacy of GS inhibition at 10 and 20 mg/kg only. Glufosinate reduced glutamine levels in tumors (Fig 3C) and plasma (Fig EV2H) in a dose-dependent manner,

without showing any effect on GS expression (Fig EV2I). Although tumor volumes and weights were unchanged (Figs 3D and EV2J), metastases in glufosinate-treated versus untreated mice were significantly decreased (Fig 3E). We then characterized the features of the macrophage infiltrate at the tumor and metastatic sites by analyzing expression of the M1 marker CD11c and the M2-like marker CD206 (MRC1) in F4/80$^+$ cells (Rolny et al, 2011). Although overall F4/80$^+$ macrophage infiltration in the tumor or in the metastasis was comparable in treated versus untreated mice (Fig EV2K and L), we confirmed that both TAMs and MAMs in treated LLC tumor-bearing mice were prevalently CD11c$^{high}$ (Fig 3F and H) and that TAMs in treated LLC tumor-bearing mice were prevalently CD206$^{low}$ (Fig 3G). Furthermore, qPCR analysis showed that TAMs sorted

from glufosinate-treated mice displayed a lower expression of the M2-specific marker Ccl22, Arg1, and Ccl17 (Fig 3I), and an increased expression of the M1-specific markers Tnfa, Cxcl9, Nos2, Cd86, and Cd80 (Fig 3J). Overall, these data indicate that glufosinate skews both TAMs and MAMs away from the M2-like phenotype in favor of more M1-like features.

We also investigated the tumor vasculature and T cells since a shift in macrophage phenotype should impinge on angiogenesis and adaptive immunity (Rolny et al, 2011). Indeed, both blood vessel length and area were reduced in glufosinate-treated versus untreated tumors (Fig 4A and B). However, tumor vessels in glufosinate-treated mice displayed increased vascular coverage and perfusion as shown, respectively, by an increased expression of α-smooth

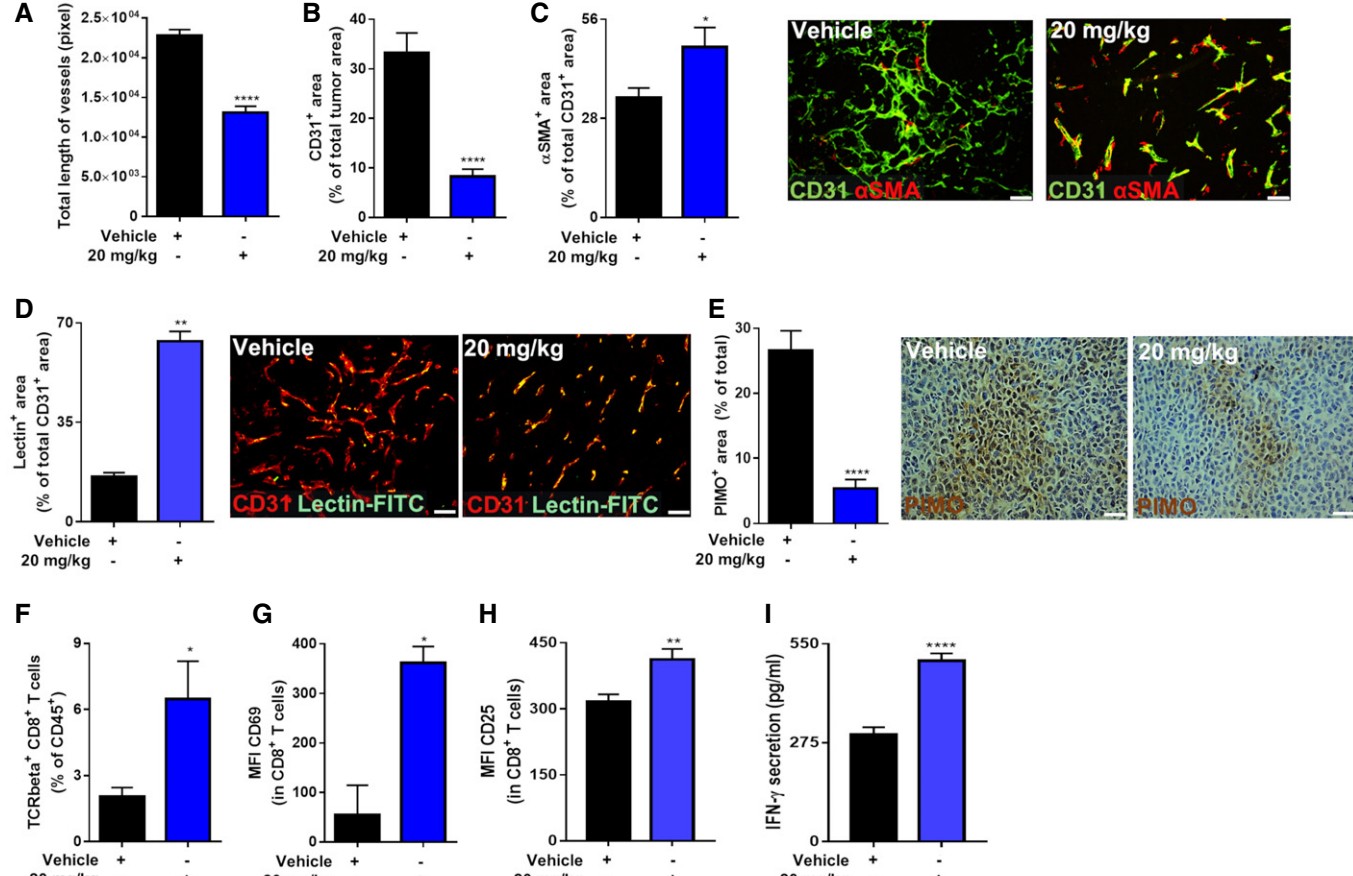

**Figure 4. Glufosinate treatment induces CTL accumulation and tumor vessel normalization.**

A–C   Quantification of total length of vessels (A), of CD31$^+$ tumor vessel area (B), and of αSMA$^+$ pericyte-covered vessels over the total number of CD31$^+$ vessels (with representative images) (C) in vehicle and glufosinate (20 mg/kg)-treated mice (n = 6).

D   Quantification and representative images of lectin$^+$ tumor vessel area over the total number of CD31$^+$ vessels in vehicle and glufosinate (20 mg/kg)-treated mice (n = 6).

E   Quantification and representative images of PIMO$^+$ tumor hypoxic areas in vehicle and glufosinate (20 mg/kg)-treated mice (n = 6).

F–H   FACS quantification of TCRbeta$^+$ CD8$^+$ cytotoxic T cells in tumors of vehicle and glufosinate-treated mice (F). Evaluation of CD69 (G) and CD25 (H) proteins, expressed as mean fluorescence intensity (MFI), was determined by flow cytometry on the surface of responders CD8$^+$ T cells (n = 4).

I   Quantification of secreted IFNγ in interstitial tumoral fluid of vehicle and glufosinate-treated mice (n = 6).

Data information: In each histological quantification, n represents the number of animals. Six images per tumor were analyzed. Scale bars: 50 μm. Data are reported as means ± SEM. *P < 0.05, **P < 0.01, ***P < 0.001, ****P < 0.0001. Exact P values and statistical tests are reported for each experiment in Appendix Table S2.
Source data are available online for this figure.

muscle actin (α-SMA) (Fig 4C), increased lectin perfusion (Fig 4D), and reduced tumor hypoxia (Fig 4E), all signs of tumor vessel normalization (Mazzone et al, 2009).

When quantifying T-cell infiltration by fluorescence-activated cell sorting (FACS) in glufosinate-treated versus untreated mice, we observed that glufosinate enhanced the infiltration and the activation of CD8$^+$ T cells (Fig 4F) as indicated by a significant up-regulation of CD69 and CD25 expression (Fig 4G and H). Furthermore, intratumoral IFNγ was significantly higher in glufosinate-treated versus

untreated mice (Fig 4I). In contrast, intratumor infiltration of CD4$^+$ T cells did not change (Fig EV2M). Altogether, these data demonstrate that glufosinate treatment in LLC-bearing mice reduces metastasis by hindering the angiogenic, immunosuppressive, and pro-metastatic potential of TAMs. A similar effect on tumor growth and metastasis was obtained when C57Bl/6N mice were treated with the ammonium salt of glufosinate (Fig EV2N and O).

To study and dissect the effect of glufosinate on cancer cell spreading and growth in the presence of a primary tumor

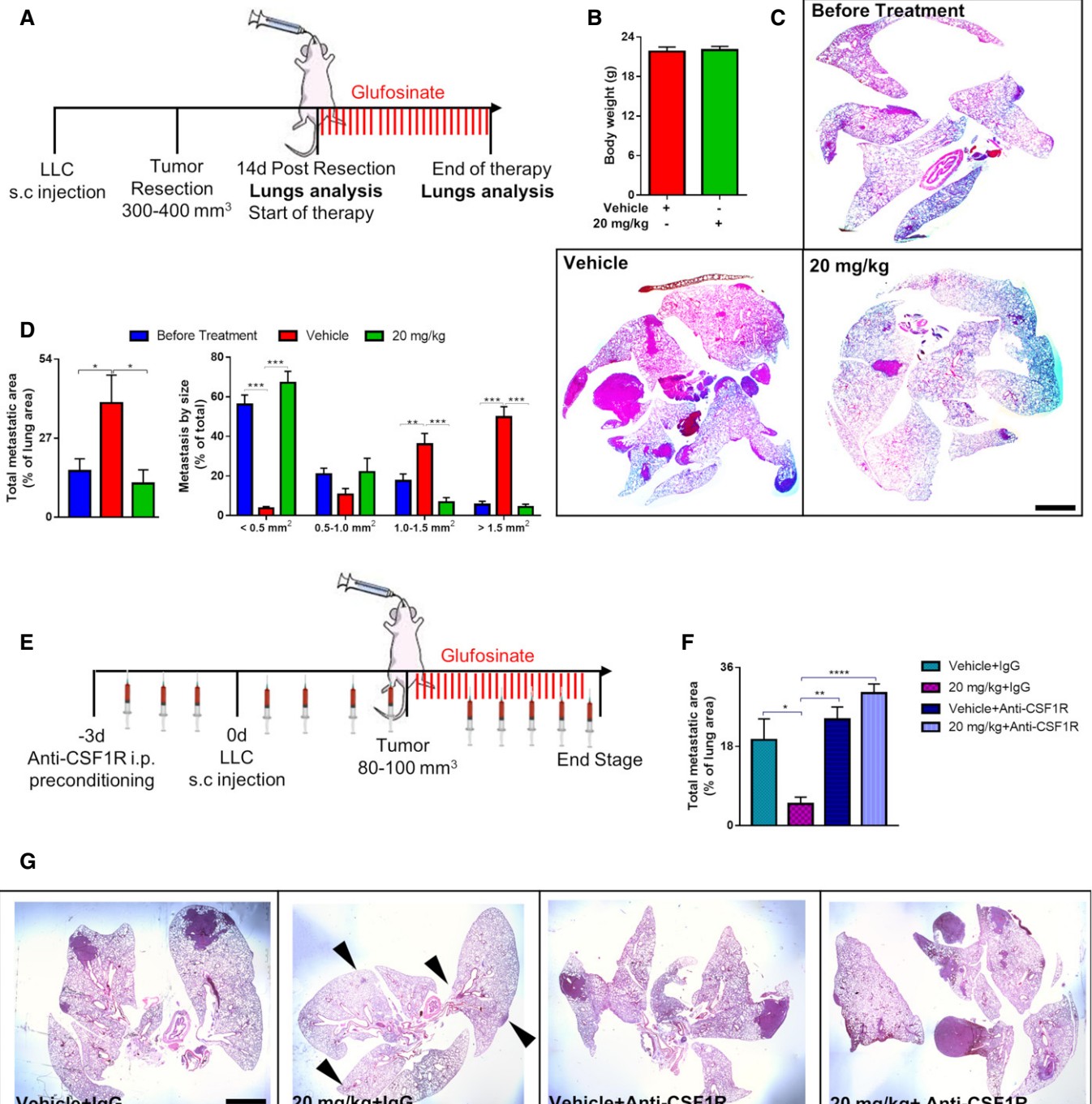

**Figure 5.**

**Figure 5. Glufosinate impairs metastatic growth after LLC tumor resection.**

A  Experimental plan. Mice were subcutaneously injected with LLC cells. When the tumor reached 300–400 mm³ size, mice were tumor resected. 14 days after resection, mice were treated by gavage with glufosinate (20 mg/kg) for 14 days.

B  Evaluation of body weight in vehicle and glufosinate (20 mg/kg)-treated mice (n = 8).

C  Representative images of metastasis at the beginning and end of therapy. Evaluation of the metastatic burden following glufosinate treatment by H&E staining (n = 8).

D  Evaluation of lung total metastatic area and of lung metastatic lesions per size by H&E staining in vehicle and glufosinate (20 mg/kg)-treated mice (n = 8).

E  Experimental plan for macrophage depletion setting. Mice were preconditioned for 3 days by i.p. injection with anti-CSF1R antibody (or isotype IgG control) before LLC (CD90.1⁺) tumor cell injection and then injected 3 times a week until sacrifice. When the tumor reached 80–100 mm³, mice were treated by gavage with vehicle or glufosinate (20 mg/kg) for 14 days.

F, G  Quantification (F) and representative images (G) of total metastatic area in LLC tumor-bearing mice preconditioned with IgG or anti-CSF1R antibody and treated with vehicle or glufosinate (20 mg/kg).

Data information: In each histological quantification, n represents the number of animals. Six images per lung were analyzed. Scale bars: 2 mm. Data are reported as means ± SEM. *P < 0.05, **P < 0.01, ***P < 0.001, ****P < 0.0001. Exact P values and statistical tests are reported for each experiment in Appendix Table S2.
Source data are available online for this figure.

(synchronous metastasis) compared to the effect of glufosinate on disease relapse and metastatic growth in the absence of a primary tumor (metachronous metastasis), LLC tumors were surgically resected (at an average size of 300–400 mm³) 10 days after subcutaneous implantation and mice were treated 14 days after resection with glufosinate (20 mg/kg) until the end stage (Fig 5A). Change in body weight following glufosinate treatment was not observed (Fig 5B). We found that the total metastatic area analyzed on H&E-stained lung sections was reduced (Fig 5C and D), with an increased incidence of metastatic lesions smaller than 0.5 mm² and a decreased incidence of those bigger than 1.0 mm² in glufosinate-treated versus untreated mice (Fig 5D). Overall, our data suggest that glufosinate hampers metastasis formation by likely preventing cancer cell intravasation, extravasation, and metastatic growth (that is a preventive mechanism of action on metastasis formation).

Since GS is expressed by other cells in the TME, glufosinate-mediated GS inhibition in these cells cannot be excluded. To study the relevance of non-macrophagic GS blockade by glufosinate in the inhibition of metastasis, we tested the effect of glufosinate in LLC-bearing mice treated with an anti-CSF1R antibody to almost completely prevent intratumor infiltration of TAMs (Fig 5E). Glufosinate did not affect tumor volume (Fig EV3A) and weight (Fig EV3B) either with or without anti-CSF1R. Anti CSF1R treatment significantly depleted TAMs (Fig EV3C and D). However, glufosinate was able to inhibit metastasis only in absence of anti-CSF1R, while TAM depletion by anti-CSF1R completely neutralized glufosinate effects on metastasis (Fig 5F and G), thus following the same trends observed on CTCs (Fig EV3E). In contrast, the effect of glufosinate on blood vessels was maintained also when TAMs were depleted (Fig EV3F and G), pointing to a direct anti-angiogenic effect on GS inhibition on endothelial cells even in absence of vessel-remodeling macrophages. These findings clearly indicate that TAM rewiring toward an M1-like phenotype, rather than TAM depletion, effectively reduces tumor metastasis. Overall, these results show that the antimetastatic effect of glufosinate occurs mainly through macrophages.

To investigate its role in preventing immune suppression and reducing the metastatic burden in a melanoma model, glufosinate was administered to CD90.2⁺ C57BL/6 mice injected orthotopically (intradermally) with YUMM1.7 melanoma cells, expressing the

**Figure 6. Glufosinate treatment promotes M1-like TAMs, CTL tumor accumulation, vessel pruning, and normalization and inhibits metastasis in a murine model of melanoma.**

A  Experimental plan. Mice were intradermally injected with YUMM 1.7 (CD90.1⁺) cells. When the tumor reached 80–100 mm³ size, mice were treated by gavage with glufosinate (20 mg/kg) or vehicle for 14 days.

B, C  Intradermal YUMM 1.7 (CD90.1⁺) tumor growth over the time (B) and number of metastatic lesions per lung (C) in vehicle and glufosinate (20 mg/kg)-treated mice (pool of 2 independent experiments; 10 mice per condition in total).

D  Representative images of metastatic lesions checked by DAB-H stain for CD90.1 positivity in vehicle and glufosinate (20 mg/kg)-treated mice (n = 6). Six images per lung were analyzed. Scale bar: 2 mm. Zoomed area scale bar: 50 μm.

E, F  FACS quantification of M1-like MHC class IIʰⁱᵍʰ (E) and MHC class IIˡᵒʷ (F) CD11c⁺ CD206⁻ TAMs over the total number of F4/80⁺ cells in vehicle and glufosinate (20 mg/kg)-treated mice (n = 6).

G  Quantification and representative images of CD11c⁺ cells over the total number of F4/80⁺ cells in vehicle and glufosinate (20 mg/kg)-treated mice (n = 6). Six images per tumor were analyzed. Scale bars: 50 μm.

H, I  FACS quantification of M2-like MHC class IIʰⁱᵍʰ (H) and MHC class IIˡᵒʷ (I) CD11c⁻ CD206⁺ TAMs over the total number of F4/80⁺ cells in vehicle and glufosinate (20 mg/kg)-treated mice (n = 6).

J  Quantification and representative images of CD206⁺ cells over the total number of F4/80⁺ cells in vehicle and glufosinate (20 mg/kg)-treated mice (n = 6). Six images per tumor were analyzed. Scale bars: 50 μm.

K, L  FACS quantification of M1-like MHC class IIʰⁱᵍʰ (K) and M2-like MHC class IIˡᵒʷ (L) TAMs over the total number of F4/80⁺ cells in vehicle and glufosinate (20 mg/kg)-treated mice (n = 6).

M  Quantification of MHC-II protein, expressed as mean fluorescence intensity (MFI), on F4/80⁺ cells, was obtained by flow cytometry (n = 6).

N  RT–PCR quantification of M2 (Arg1 and Cxcr4) and M1 (Cxcl9 and Nos2) markers in vehicle and glufosinate (20 mg/kg)-treated mice (n = 6).

Data information: Data are reported as means ± SEM. *P < 0.05, **P < 0.01, ***P < 0.001, ****P < 0.0001. Exact P values and statistical tests are reported for each experiment in Appendix Table S2.
Source data are available online for this figure.

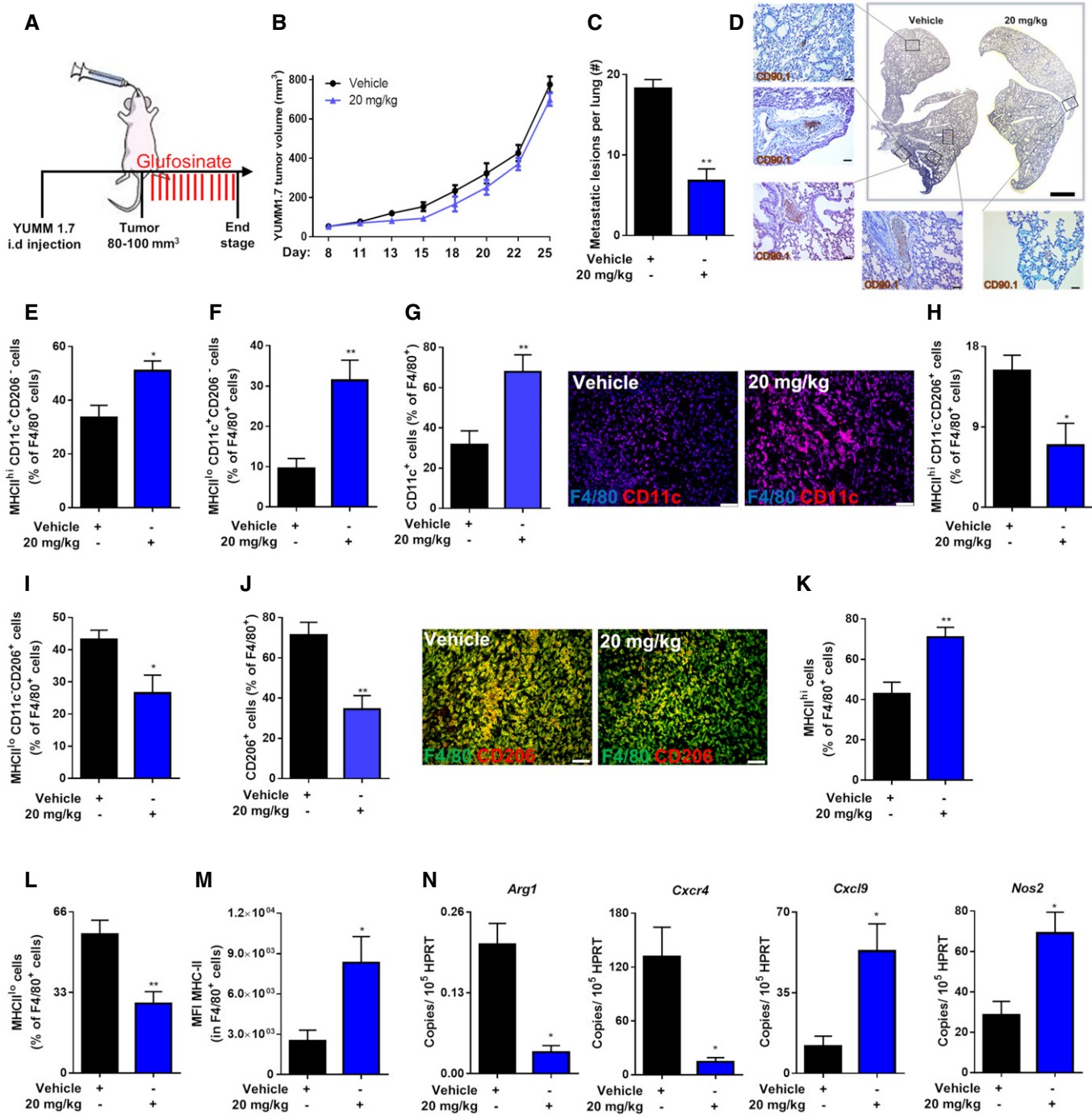

Figure 6.

syngenic marker CD90.1 (Fig 6A). In line with the LLC model, glufosinate did not affect either the average tumor volume (Fig 6B) or tumor weight (Fig EV4A). Body weights were comparable as well (Fig EV4B). We then checked the presence of lung metastatic foci, and we found a significant reduction of metastatic lesions in glufosinate-treated mice compared to the vehicle group (Fig 6C and D).

Also in this model, we evaluated the features of the F4/80[+] TAM infiltrate by FACS and histological analysis for the M1 markers CD11c or MHCII and the M2 marker CD206 (MRC1) (Pucci *et al*, 2009; Andreu *et al*, 2010; Movahedi *et al*, 2010; Rolny *et al*, 2011;

Laoui *et al*, 2014; Wenes *et al*, 2016). The overall F4/80[+] macrophage abundance in the tumors was comparable in both groups (Fig EV4C and D). However, in tumors treated with glufosinate we found an increase of M1-like macrophages (Fig 6E–G) and a reduction of M2-like TAMs as confirmed by both FACS and histological analysis (Fig 6H–J). Glufosinate also enlarged the fraction of TAMs displaying high antigen presentation potential as shown by the increased percentage of MHCII[hi] cells (Fig 6K and L) and augmented MHCII expression per cell (Fig 6M). qPCR experiments on mRNA from sorted TAMs confirmed these results, as both *Arg1* and *Cxcr4*

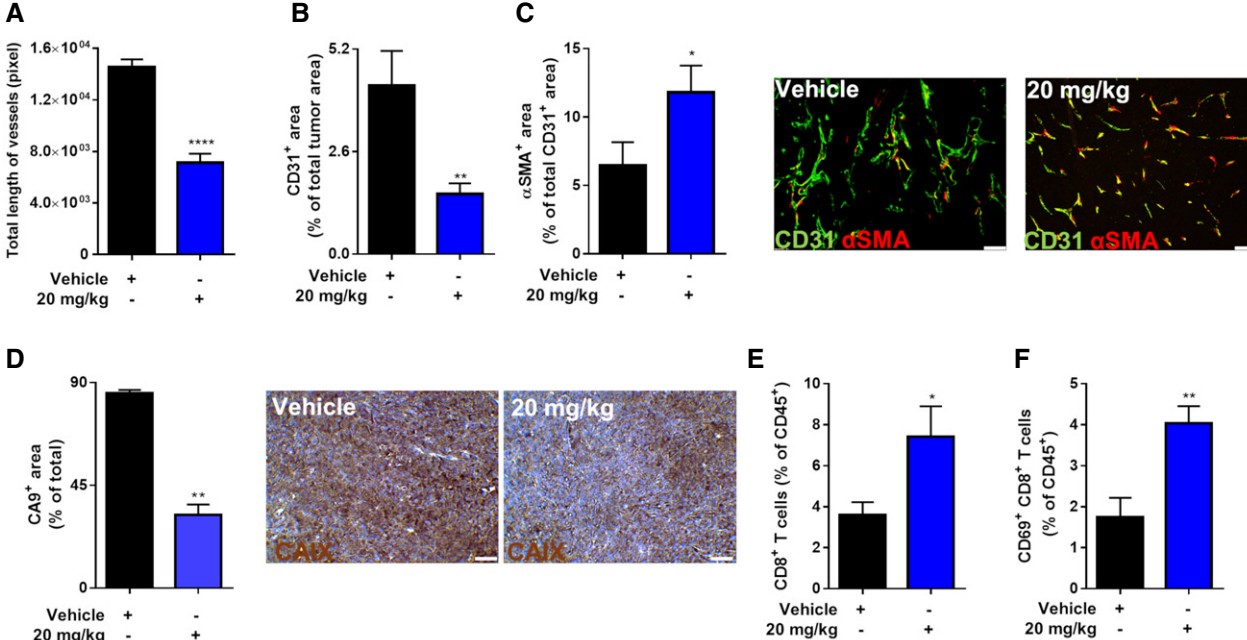

**Figure 7. Glufosinate treatment promotes CTL tumor accumulation, vessel pruning, and normalization in a murine model of melanoma.**

A–C  Quantification of total length of vessels (A), of CD31+ tumor vessel area (B), and of αSMA+ pericyte-covered vessels over the total number of CD31+ vessels (with representative images) (C) in vehicle and glufosinate (20 mg/kg)-treated mice (*n* = 6). Six images per tumor were analyzed. Scale bar: 50 μm.

D  Quantification and representative images of CA9+ tumor hypoxic areas in vehicle and glufosinate (20 mg/kg)-treated mice (*n* = 6). Six images per tumor were analyzed. Scale bar: 50 μm.

E, F  FACS quantification of CD8+ cytotoxic T cells (E) and CD69+ CD8+ T cells (F) over the total number of CD45+ in vehicle or glufosinate (20 mg/kg)-treated mice (*n* = 6).

Data information: Data are reported as means ± SEM. *$P < 0.05$, **$P < 0.01$, ***$P < 0.001$, ****$P < 0.0001$. Exact *P* values and statistical tests are reported for each experiment in Appendix Table S2.

Source data are available online for this figure.

expression was decreased, whereas *Cxcl9* and *Nos2* were increased in response to glufosinate treatment *in vivo* (Fig 6N).

In line with the results in LLC tumors, glufosinate decreased both tumor blood vessel length (Fig 7A) and area (Fig 7B) in YUMM tumors as well. Tumor vessels in glufosinate-treated mice displayed increased vascular coverage as indicated by an increased percentage of α-SMA+ peri-vascular area (Fig 7C). This associated with reduced tumor hypoxia, assessed by staining for CA9 upon glufosinate treatment (Fig 7D), altogether suggesting increase blood vessel functionality.

To confirm that GS inhibition counteracts the immunosuppressive phenotype of macrophages, we quantified by FACS intratumoral T cells as a readout. Since the effects of glufosinate on CD4+ T cells in the LLC model were negligible, we focused on CD8+ T cells, to find that CD8+ T-cell infiltration and activation were significantly increased in glufosinate-treated versus untreated mice (Fig 7E and F). These data demonstrate that glufosinate treatment in a skin cancer model reduces the size of metastatic lesions by hindering the immunosuppressive and pro-metastatic potential of TAMs.

In BALB/c mice, the highest well-tolerated dose of glufosinate was 10 mg/kg. We then assessed if 4T1 triple-negative breast tumors, orthotopically implanted in the mammary fat pad, were sensitive at this dose of glufosinate (Fig 8A). Also, in this cancer model, glufosinate did not reduce tumor volumes (Fig 8B) and did not induce weight loss in mice (Fig 8C). However, we found a

significant reduction of metastatic nodules (Fig 8D) and total metastatic area (Fig 8E) in glufosinate-treated mice compared to vehicle group. Furthermore, glufosinate significantly increased the incidence of metastatic lesions smaller than 0.5 mm² and reduced the incidence of those bigger than 1.0 mm² (Fig 8F and G). At the metastatic foci, glufosinate treatment did not induce any change in the total F4/80+ MAM infiltration (Fig 8H); however, upon treatment, MAMs were mainly CD206low (Fig 8I).

To measure the effects of glufosinate on cancer cell intravasation, the amount of circulating cancer cells was evaluated, at the time of tumor resection, in the blood of mice orthotopically injected with 4T1-GFP+ (Fig 8J). Glufosinate reduced the amount of GFP+ cells (Fig 8K). Outgrown GFP-labeled CTC colonies from cultured freshly drawn blood were significantly reduced by glufosinate (Fig 8L). In neoadjuvant therapeutic regimen, glufosinate improved the survival of post-surgical mice (Fig 8M).

To assess the effects of glufosinate on the extravasation of circulating tumor cells (CTCs) and their seeding at the metastatic site, after a 2-day preconditioning with glufosinate treatment of the host mice, 4T1-GFP+ cells were injected by tail vein to mimic the progression of metastases unrespective of the primary tumor (Fig 8N). Mice were sacrificed after 48 h of treatment, and the lungs were analyzed by qPCR. Glufosinate reduced GFP expression in lungs (Fig 8O). Altogether, glufosinate is able to impair both cancer cell intravasation and extravasation.

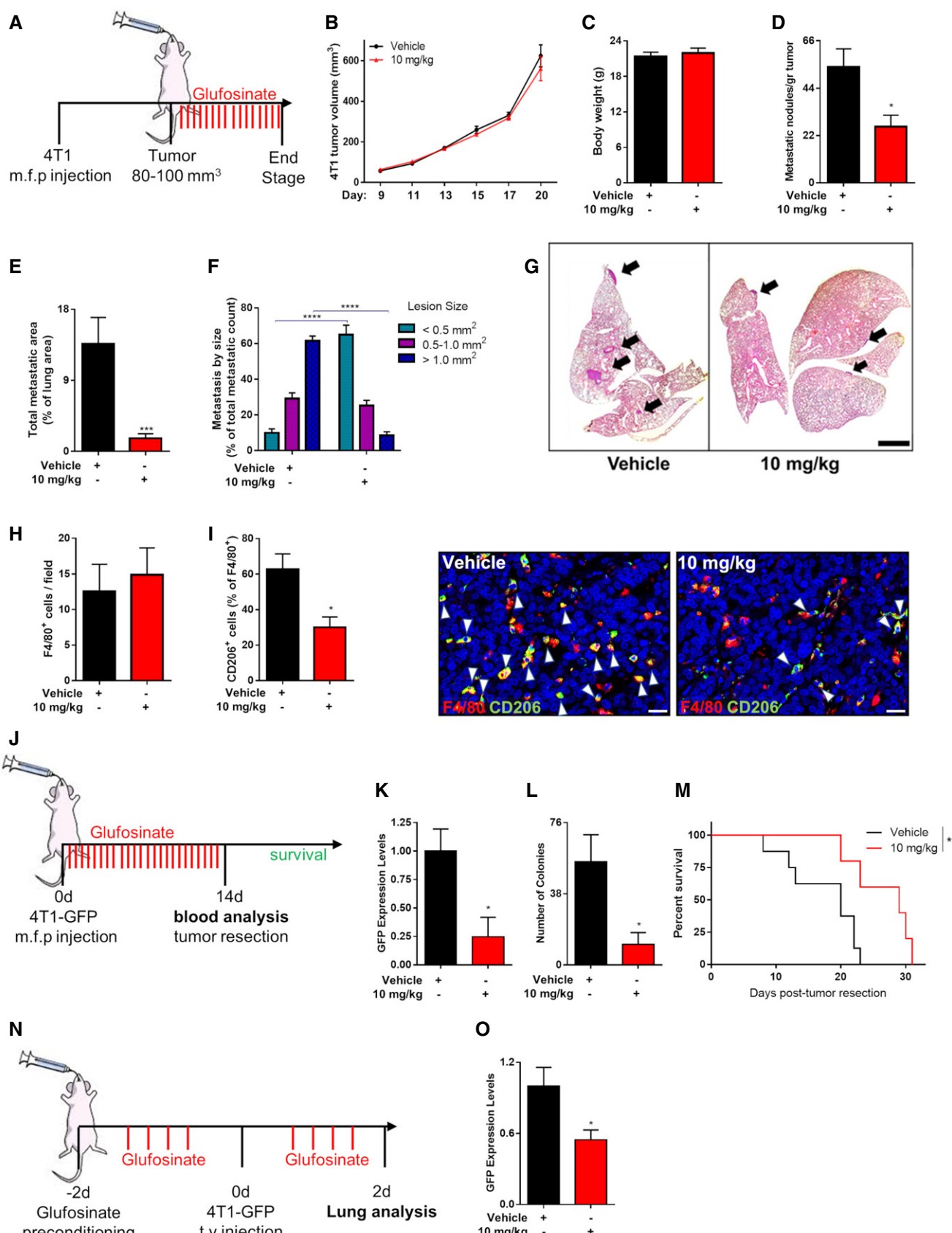

Figure 8.

◄

**Figure 8.  Glufosinate treatment inhibits metastasis in a murine model of breast cancer.**

A       Experimental plan. Mice were injected with 4T1 cells in the mammary fat pad. When the tumor reached 80–100 mm³ size, mice were treated by gavage with glufosinate (10 mg/kg) or vehicle for 14 days.

B–D    Evaluation of 4T1 tumor growth over the time (B), mice body weight (C), and number of metastatic nodules (D), in vehicle and glufosinate (10 mg/kg)-treated mice (pool of 2 independent experiments; 10 mice per condition in total).

E–G    Total metastatic area (E) and number of lung metastatic lesions per size by H&E staining (F) with representative images (G) in vehicle and glufosinate (10 mg/kg)-treated mice (n = 8). Six images per lung were analyzed. Scale bar: 2 mm.

H, I    Quantification of F4/80⁺ macrophage density (H), of F4/80⁺ CD206⁺ cells infiltration in the lung metastasis and representative images (I), in vehicle and glufosinate-treated mice (n = 6). Six images per lung were analyzed. Scale bar: 20 μm.

J       Experimental plan for intravasation setting. Mice were injected with 4T1-GFP⁺ cells in the mammary fat pad and immediately treated by gavage with glufosinate (10 mg/kg) or vehicle for 14 days (neoadjuvant therapeutic regimen). After 14 days of treatment, at the time of tumor resection, blood was collected and analyzed for GFP⁺ circulating tumor cells (CTCs). Survival studies were performed until the natural death of the post-surgical mice.

K–M    GFP⁺ CTCs amount in blood, expressed as GFP expression levels, was determined by qPCR (n = 8) (K). Blood was drawn from treated and untreated tumor-bearing mice at the time of tumor resection and plated in culture dishes. The formation of tumor cell colonies was traced over time. Graph shows the number of GFP⁺ tumor cell colonies (n = 8) (L). After treatment and tumor resection, overall survival studies were performed until the natural death of the post-surgical mice (n = 11) (M).

N       Experimental plan for extravasation setting. GFP-expressing 4T1 were injected intravenously into BALB/c mice preconditioned with glufosinate (10 mg/kg) or vehicle for 2 days. After 2 days of treatment, the overall lung metastatic breast cancer cell burden was determined.

O       GFP⁺ cancer cells amount in lungs, expressed as GFP expression levels, was determined by qPCR in vehicle and glufosinate (10 mg/kg)-treated mice (n = 8). The results were normalized to the vehicle group.

Data information: Data are reported as means ± SEM. *P < 0.05, **P < 0.01, ***P < 0.001, ****P < 0.0001. Exact P values and statistical tests are reported for each experiment in Appendix Table S2.

Source data are available online for this figure.

## Glufosinate distributes in tissues but does not induce liver or brain toxicity

Having established that glufosinate produces a significant effect in different cancer-bearing mice with respect to angiogenesis, immune suppression, and metastasis formation, we needed to assess the amount of the molecule actually found in non-malignant tissues, together with the evaluation of the main hematological parameters as well as brain and liver functions.

Treatment with glufosinate did not induce neither weight loss in mice (Fig 9A) nor it modified blood counts (Fig 9B). Although liver accumulation of glufosinate was significant in treated versus untreated mice, with the highest amount of glufosinate detected when the 20 mg/kg dose was administered (Fig 9C), the blood indicators of liver function were unchanged (Fig 9D), suggesting the glufosinate is not hepatotoxic at the used concentrations. Finally, glufosinate in the brain was mildly accumulating though its concentration was higher, though not significantly, at 20 mg/kg versus 10 mg/kg (Fig 9E). We speculate that this low but dose-dependent accumulation is due to the presence of glufosinate in the bloodstream of the brain favoring the idea that glufosinate has not high propensity to cross the blood–brain barrier (Kishore & Shah, 1988). This idea was further corroborated when performing brain toxicity assays with respect to both sensory neurons and motoneurons. To this end, we measured the grip strength (Fig 9F and G) and the latency to fall off an accelerating rotarod (Fig 9H) in vehicle and glufosinate-treated mice before the treatment (baseline), after 1 day (acute toxicity) and after 15 days (chronic toxicity). Glufosinate treatment did not alter the mean paw withdrawal force measured by the von Frey test, excluding the presence of tactile allodynia (Fig 9I and J). All together, these results demonstrate that glufosinate treatment accumulates not only in pathological tissues but also in the normal tissues, it barely crosses the blood–brain barrier and it does not induce liver and moto/sensory neuron toxicity as well as blood count defects.

## Discussion

A tumor consists not only on transformed cells, but also on stromal cells that can be recruited and hijacked by cancer cells. It follows that the interactive mechanisms between different cells within the tumor, that define the so-called tumor microenvironment (TME), are also very complex and importantly contribute to tumor malignancy (Binnewies et al, 2018). An important stromal component in the TME is represented by TAMs (Mazzone et al, 2018; Prenen & Mazzone, 2019). TAMs take part to several steps in the formation of metastasis (Pollard, 2004; Komohara et al, 2016). In particular, these cells help cancer cells evading the immune system and promote angiogenesis. Additionally, the presence of macrophages at the pre-metastatic niche supports a role for these cells in preparing the metastatic niche for the arrival of newly disseminated cancer cells or in favoring cancer cell survival in a site different from the primary tumor (Nielsen & Schmid, 2017). During these different steps, cancer cells adapt their metabolism to increase energy production to proliferate or to allow survival (Renner et al, 2017). In doing so, these cells modify the metabolic composition of the TME, inducing the recruitment of macrophages through mechanisms that can be mediated by functionally relevant metabolic reprogramming. Identification of the metabolic checkpoints regulating macrophagic function, which can be targeted to improve cancer specific immune responses, is now a promising strategy for therapeutical intervention (Mazzone et al, 2018). We have recently identified GS as an enzyme playing a fundamental role in the acquisition of the protumoral and pro-metastatic function of TAMs (Palmieri et al, 2017b). These results were achieved by pharmacological GS inhibition in vitro and by genetic GS ablation specifically in macrophages in a single murine model of cancer (Palmieri et al, 2017b). However, the efficacy of in vivo pharmacological targeting of GS in several tumor model has never been proven so far.

Here we present data on the pharmacological inhibition of GS in lung, breast, and skin murine models of metastatic cancer by using

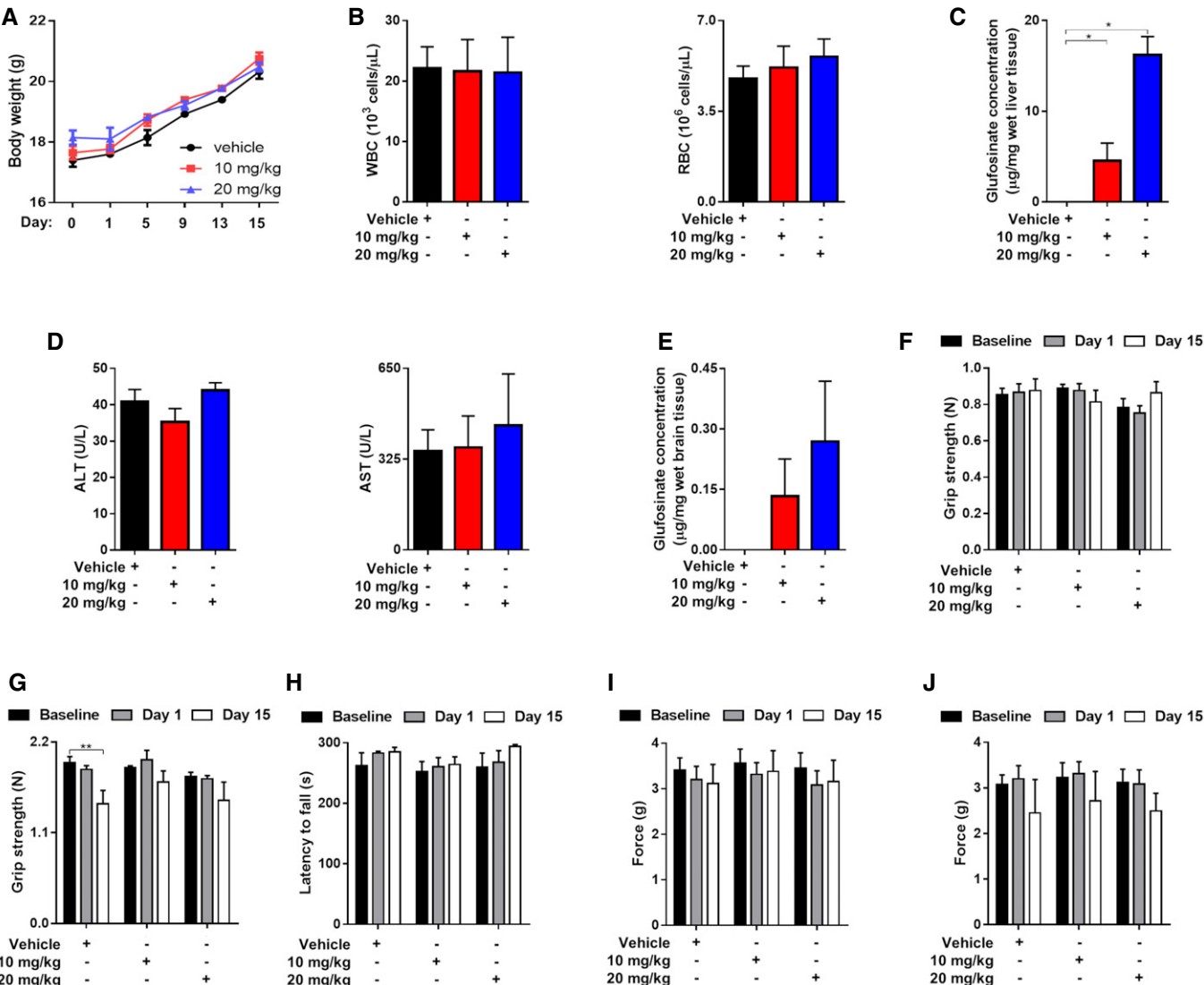

**Figure 9. Glufosinate treatment does not display blood, liver and neuronal toxicity.**

A   Time course evaluation of murine body weight during treatment in vehicle and glufosinate (10 and 20 mg/kg)-treated mice (pool of 2 independent experiments; 10 mice per condition in total).

B   Blood counts of WBC and RBC in vehicle and glufosinate (10 and 20 mg/kg)-treated mice ($n = 8$).

C, D   Evaluation of liver toxicity following glufosinate treatment. Quantification of liver glufosinate (C) and of hepatotoxicity markers ALT (alanine aminotransferase) and AST (aspartate aminotransferase) (D) in serum of vehicle and glufosinate (10 and 20 mg/kg)-treated mice ($n = 8$).

E–J   Evaluation of neuronal toxicity following glufosinate treatment. Quantification of brain glufosinate (E), of the grip strength for front paws (triangular bar) (F), for all paws (rectangle grid) (G), of the latency to fall off an accelerating rotarod (H), and of the mean paw withdrawal force that caused animals' response in the von Frey test (left hindpaw, (I); right hindpaw, (J)) as an indicator of the development of tactile allodynia in vehicle and glufosinate-treated mice ($n = 4$), before the treatment (baseline), after 1 day (acute toxicity), and after 15 days (chronic toxicity) (pool of 2 independent experiments; 10 mice per condition in total).

Data information: Data are reported as means $\pm$ SEM. *$P < 0.05$, **$P < 0.01$, ***$P < 0.001$, ****$P < 0.0001$. Exact $P$ values and statistical tests are reported for each experiment in Appendix Table S2.

glufosinate, a known inhibitor of plant GS (Occhipinti *et al*, 2010). Glufosinate treatment *in vitro*, as low as 10 μM, recapitulates the effect of MSO on IL10 macrophages, leading to a metabolic reprogramming, associated with a HIF1α-dependent M1-like phenotype, that functionally translates into a decreased immune suppression, angiogenesis, and cancer cell invasion.

Our findings that GS-inhibited macrophages counter T-cell immunosuppression by engaging HIF1α seem in sharp contrast with

the protumoral, immunosuppressive role of HIF1α in hypoxic TAMs (Doedens *et al*, 2010). We show that GS inhibition maintains mTOR active even in hypoxic conditions. This occurs through downregulation of REDD1, a known negative regulator of mTOR in hypoxia. Active mTOR in hypoxia shields macrophages from the immunosuppressive function dictated by hypoxia, forcing macrophages to display the classical functional program of normoxic pro-inflammatory M1-like macrophages, wherein mTOR is

physiologically active. Concomitant activation of HIF1α and mTOR, and in particular TORC1 (Covarrubias *et al*, 2015), is responsible for the acquisition of an immunostimulatory and antitumoral macrophage phenotype. Our results clearly demonstrate that hypoxic TAMs can engage in a pseudo-normoxic state, which is triggered by GS inhibition.

Our *in vitro* data are confirmed in several mouse models of cancer, in which glufosinate treatment significantly reduces metastasis formation and promotes a phenotype switch of TAMs toward a M1-like state, similarly to what observed in the conditional, macrophage-specific, GS knockout mouse (Palmieri *et al*, 2017b). Since glufosinate might reach all the cells of the TME, it cannot be ruled out that stromal cellular components other than TAMs might contribute to the antimetastatic effect displayed by glufosinate. Indeed genetic targeting of GS in cancer-associated fibroblasts (CAFs) in a murine model of ovarian cancer induces tumor regression (Yang *et al*, 2016). Additionally, GS inhibition in endothelial cells might impair vessel sprouting during vascular development (Eelen *et al*, 2018). However, TAM depletion in glufosinate-treated tumor-bearing mice completely prevents the antimetastatic effect of GS inhibition, arguing that the drop in metastatic burden in glufosinate-treated tumor-bearing mice is greatly due to the revert of the immunosuppressive and proangiogenic function of TAMs. Yet, the anti-angiogenic effect of glufosinate remains unaltered even upon TAM depletion, suggesting an endothelial cell autonomous mechanism of inhibition (that is macrophage-independent) of glufosinate on tumor blood vessel growth (Eelen *et al*, 2018). The alleged effect of glufosinate on endothelial cells is sufficient to prune tumor blood vessels but not to reduce metastasis. This is consistent with previous findings by our laboratory and others (Shojaei *et al*, 2007; Casazza *et al*, 2012), altogether showing that LLC-derived primary tumors and metastases are refractory to anti-angiogenic therapies because of compensatory responses elicited by myeloid cells, which contribute in several ways to angiogenesis (*i.e.*, by modulating both endothelial cells and mural cells) and also impose their control on the adaptive immune system.

The finding, confirmed in several tumor models, that glufosinate-induced rewiring of the TAM phenotype toward an M1-like state is associated with the inhibition of metastases but not of the primary tumor points to the involvement of TAMs in the antimetastatic effect displayed by glufosinate. Indeed, TAMs are known to affect distant dissemination of cancer cells more than tumor growth (Qian *et al*, 2015). TAMs can also promote tumor vascular abnormalities; therefore, the ability of restoring "tumor vessel normalization" results in reduced hypoxia and leakage, as well as increased delivery of therapeutic drugs to the tumor (Mazzone *et al*, 2009; Rolny *et al*, 2011; Leite de Oliveira *et al*, 2012), and increased immunosurveillance that can restrain metastatic outgrowth (Eyles *et al*, 2010). Based on the effects of glufosinate in rewiring the TAM phenotype toward a M1-like state, each of the described mechanisms might conceivably play a role in inhibiting metastatic growth. The additional effect of glufosinate in increasing tumor vessel perfusion suggests that this drug might be exploited not only for its immunomodulatory properties, but also in combination with state-of-the-art anti-cancer drugs (i.e., chemotherapy) to enhance their delivery and therapeutic efficacy against both primary tumor and metastasis. Importantly, macrophages patrolling the metastatic niche (namely MAMs, metastasis-associated macrophages) have been involved in cancer cell extravasation, angiogenesis, as well as cancer cell survival and proliferation (Chen *et al*, 2011; Qian *et al*, 2011; Nielsen *et al*, 2016; Celus *et al*, 2017). By following experimental metastases and metastatic growth upon primary tumor resection (which mimic settings of metachronous metastases), we prove that MAMs are also governed by GS in their M2 phenotype and its inhibition is favouring a switch toward the M1-like phenotype. This favors the use of glufosinate in controlling a metastatic relapse. However, we went beyond these findings, suggesting that glufosinate limits growth of established metastatic lesions, overall opening the possibility to use this type of therapeutic approach in a curative setting, following disease relapse or in an adjuvant setting (following the resection of the primary tumor). From a metabolic point of view, since MAMs are known to promote metastasis formation through signaling and metabolic conditioning of the pre-metastatic niche to prepare the microenvironment for cancer cells survival and proliferation (Qian *et al*, 2015; Celus *et al*, 2017; Prenen & Mazzone, 2019), it can be speculated that GS expression in MAMs might be fundamental to supply glutamine, which is probably a limiting metabolite in the delicate step of metastatic growth.

Exploitation of targeted therapies against TAMs represents a promising option to fight cancer. Indeed some clinical efficacy has been shown by depleting TAMs or by inhibiting their protumoral functions, for instance, by blocking the CSF1R signaling (Ries *et al*, 2014). However, general ablation of protumoral macrophage functions or of total macrophages might not represent the correct strategy due to the possible systemic harmful consequences. Identification of metabolic checkpoints of macrophage function might represent a promising strategy to induce selective reprogramming of abundant protumoral macrophages toward an antitumoral phenotype, a process now known a TAM re-education (Beatty *et al*, 2011; Rolny *et al*, 2011; Casazza *et al*, 2013). Furthermore, new drugs based on small organic compounds active as enzyme inhibitors, instead of antibodies, might be available at lower costs. Our results from glufosinate treatment demonstrate that targeting metabolic activities crucial for the acquisition of protumoral macrophage phenotypes is a safe and effective strategy against the immunosuppressive function of macrophages.

In conclusion, our data confirm that GS is a metabolic checkpoint crucial for the targeting of immunosuppressive and proangiogenic TAMs, that is ultimately relevant for metastasis formation in a synchronous setting and in a more clinically relevant metachronous setting. Overall, we show that two different doses of the GS inhibitor glufosinate do not display overt toxic effects. Our results clearly point at metabolic immunotherapy as a powerful and feasible strategy to rewire macrophage function toward an antimetastatic phenotype, which is currently strongly awaited in the fight against cancer.

# Materials and Methods

### Sequence search and analysis

Protein and genomic databases (www.ncbi.nlm.nih.gov) were screened by using the human GS (accession number NP_001028216) as a query sequence.

## Construction of expression plasmids

The coding sequence of human GS (accession numbers NM_001033044.3) was amplified by PCR from human liver cDNA. The oligonucleotide primers were synthesized corresponding to the extremities of the coding sequence, with additional BamHI and HindIII restriction sites as linkers at 5′ end allowing unidirectional cloning. The amplified product was digested (BamHI/HindIII) and cloned into the pRUN vector for expression in *E. coli*. The plasmid containing the coding sequence for human GS (hGS) was transformed into *E. coli* DH5 cells. Transformants were selected on 2× YT plates containing ampicillin (100 μg/ml) and screened by direct colony PCR and by restriction digestion of purified plasmids. The sequence of the insert was verified by DNA sequencing (StarSEQ).

## Bacterial expression and purification of recombinant GS

The expression of recombinant hGS was carried out at 37°C in *E. coli* BL21 (DE3) cells. The inclusion bodies were purified on a sucrose density gradient (Frelin *et al*, 2012; Zallot *et al*, 2013) and washed with TE buffer (10 mM Tris–HCl, 0.1 mM EDTA, pH 7.2).

## Reconstitution of functional GS and enzymatic assay

The recombinant hGS was solubilized in a solubilization buffer containing Urea (2 M), DTE (10 mM), and Tris–HCl (100 mM, pH 9.0) at 25°C under stirring for 3 or 4 h. The insoluble residues were removed by centrifugation (20,800 $g$ for 10 min at 4°C). The reaction catalyzed by hGS was measured by a microtiter assay using inorganic phosphate detection (Gawronski & Benson, 2004). The reaction mix contained 50 mM HEPES (pH 7.2), 50 mM $MgCl_2 \cdot 6H_2O$, 20 mM monosodium glutamate, 10 mM $NH_4Cl$, and purified recombinant hGS.

The final volume of reaction mix was 200 μl. To determine the amount of produced phosphate, 10 μl of each dilution of the phosphate standard from 0 to 15 mM was added to reaction mix instead of the initiation substrate. After 8 min, 50 μl of the reaction mix was added in a 96-well Microwell plate (Nunc, Roskilde, Denmark) to 150 μl of two part of 12% w/v L-ascorbic acid in 1N HCl and one part of 2% w/v ammonium molybdate tetrahydrate in $ddH_2O$, prepared just before use. After 5 min, 150 μl of 2% sodium citrate tribasic dihydrate and 1% acetic acid in $ddH_2O$ was added to the wells to stop color development. The reaction was allowed for 5 min at room temperature before reading the absorbance at 750 nm with a GloMax® Discover Multimode Microplate Reader (Promega). To determine the apparent kinetic parameters of recombinant hGS, the initial reaction activity was measured varying the concentrations of a substrate in the presence of a fixed saturating concentration of the other substrates. A series of dilutions of glutamate or ATP was prepared to determine their respective Michaelis–Menten (half-saturation) constant ($K_m$) of the human recombinant GS and the $K_i$ values of MSO or glufosinate. The GS biosynthetic reaction was also performed in presence of different concentrations of MSO or glufosinate. These compounds were added simultaneously with the other mix components before of ATP addition.

## Protein sequence sampling

GS sequences from *H. sapiens* (NP_002056.2) and *Z. mays* (NP_001105443.2) were collected from the RefSeq protein sequence database and used as queries to search for homologous sequences in *H. sapiens* (taxid:9606) (and other mammalia (taxid:40674)) and *Z. mays* (taxid:4577) (and other plant species (taxid:3193)).

## Crystal structure sampling via folding recognition

Protein-crystallized structures homologous to GS were searched by pGenThreader (Lobley *et al*, 2009) and iTasser (Yang & Zhang, 2015). A previously described method (Trisolini *et al*, 2019) allowed sampling 17 crystal structures, that were subsequently aligned and superimposed to be compared with PyMOL (Ordog, 2008). Residues within 4 Å from substrates/cofactors crystallized in the sampled structures were highlighted to compare the related ligand-binding regions, which were superimposed by using PyMOL as previously described (Pierri *et al*, 2010; Bossis *et al*, 2014; Trisolini *et al*, 2019).

## Multiple sequence analysis

The parameters selected to retain the sampled sequences were E-values, query coverage, and percentage of identical amino acids and the threshold was set at E = $10^{-86}$, query coverage = 60%, and the percentage of identical amino acids = 50. ClustalW, implemented in the sequence editor suit Jalview, was used to build a MSA of 36 sequences (10 mammalian, 9 plant, and 17 sequences from the crystallized structures) as previously indicated (Pierri *et al*, 2010; Bossis *et al*, 2014; Trisolini *et al*, 2019). MatGAT was used to calculate the amino acid identity or similarity (expressed in %) among the sequences sampled through the blast and the fold recognition tools (Campanella *et al*, 2003).

## Cell culture

Human lung carcinoma A549, murine 4T1 (GFP$^+$) mammary carcinoma, YUMM1.7 (CD90.1$^+$) melanoma, and Lewis lung carcinoma (CD90.1$^+$) cells, obtained from ATCC, were maintained in DMEM media containing 25 mM glucose, 4 mM glutamine, 100 U/ml of penicillin, 100 μg/ml of streptomycin, and 10% fetal calf serum. Cells were cultured in 5% $CO_2$/95% air at 37°C in a humidified chamber, split every 2–3 days, passaged at ratios that ranged from 1:4 to 1:8, and used up to passage 10 (Meeth *et al*, 2016). For tumor inoculation, cancer cells were harvested at approximately 60–85% confluence on the day of injection. Cells were trypsinized with 0.25% trypsin for approximately 2–3 min before deactivation with media containing 10% serum. They were then washed twice with sterile 1× PBS and counted with a hemocytometer. Cell viability was determined after 72 h of incubation, using the CellTiter 96® Non-Radioactive Cell Proliferation Assay (Promega) as described (Menga *et al*, 2017; Palmieri *et al*, 2017b).

## Cell isolation and culture

Human monocytes were isolated from human buffy coats with CD14 MicroBeads (Miltenyi Biotec Inc.) as previously described (Palmieri *et al*, 2015, 2017b). After differentiation, macrophages

were stimulated with LPS/IFNγ (for M1 polarization) and IL10 (for M2 polarization). Experiments of inhibition were performed by adding 1 mM MSO, or 10, or 20 μM glufosinate one h before the cytokines for activation. For HIF1α inhibition, cells were stimulated with LPS/IFNγ or IL10 with and/or without a previous 2 h incubation with glufosinate and 5 μM Acriflavine (ACF). Murine bone marrow-derived macrophages (BMDMs) were derived from bone marrow precursors as described before (Palmieri *et al*, 2017b). Briefly, bone marrow cells (1.6 × 10^6 cells/ml) were cultured in DMEM supplemented with 20% FBS and 30% L929 conditioned medium as a source of M-CSF. After 3 days of culture, an additional 3 ml of differentiation medium was added. At day 7, macrophages were pretreated with glufosinate, polarized with 10 ng/ml IL10 or with 100 ng/ml LPS plus 20 ng/ml IFNγ for 24 h, and finally harvested with ice-cold $Ca^{2+}$ and $Mg^{2+}$-free PBS. For the nutrient-sensitive mTOR complex 1 (mTORC1) inhibition, BMDMs were stimulated with IL10 with or without a previous 2 h incubation with glufosinate and/or 20 nM rapamycin (Rapa) (Byles *et al*, 2013) in normoxic and hypoxic conditions.

### RNA and protein expression analysis

Total RNA was extracted from cells and reverse transcribed ad described (Iacobazzi *et al*, 2009; Palmieri *et al*, 2017a,b). TaqMan probes were purchased from IDT and Thermo Fisher. Analysis was carried out with a QuantStudio real-time PCR system (Thermo Fisher). All transcript levels were normalized against the β-actin expression level.

For Western blotting analysis, whole cell lysates were prepared by treating pelleted cells with ice-cold RIPA buffer (1% Nonidet P-40, 50 mM Tris–HCl pH 7.4, 150 mM NaCl, 0.1% SDS, 2 mM EDTA, 0.5% sodium deoxycholate) containing 1 × protease inhibitors (Sigma P8340) and 1 mM PMSF (Sigma P7626) for 30 min at 4°C (Palmieri *et al*, 2017a). Proteins were analyzed by SDS–PAGE TGX Stain-Free™ Fast-Cast™ (Bio-Rad) and visualized by ChemiDoc™ MP System (Bio-Rad) or with Coomassie Blue dye (Lauderback *et al*, 2003; Palmieri *et al*, 2014, 2017b). In the case of GS, N-terminal sequencing was carried out as described previously (Todisco *et al*, 2006, 2014). The amount of purified GS was estimated by laser densitometry of stained samples, using bovine albumin as protein standard. Proteins were electroblotted with iBlot® Transfer System (Life Technologies) and subsequently treated with the different antibodies as follows: anti-GS (Sigma MAB302) (Palmieri *et al*, 2014), anti-phospho-4E-BP1 (Cell Signalling, 2855), anti-4E-BP1 (Cell Signalling, 4923), anti-p70 S6 Kinase (Cell Signalling, 9202), anti-phospho-p70 S6 Kinase (Cell Signalling, 9206), anti-REDD1 (Santa Cruz, sc-46034), anti-S6 (Cell Signalling, 2217), anti-phospho-S6 (Cell Signalling, 4858) (Wenes *et al*, 2016; DiConza *et al*, 2017), CCL18 (Novus, NBP1-79940), TNFα (Abcam, ab9635), and vinculin (Sigma V9131-.2ML), which was used as housekeeping gene. The immunoreaction was detected by the ECL plus system (Amersham). More details about reagents and dilutions are reported in Appendix Table S1.

### Mass spectrometry analysis

For mass spectrometry analysis, cells were scraped in 80% methanol to extract polar metabolites, which was subsequently centrifuged and dried using a vacuum concentrator (Palmieri *et al*, 2014, 2017a). For plasma extraction, 20 μl of medium was added to methanol and treated as above. The dried metabolite samples were stored at −80°C. For glufosinate quantification in brain, liver, and tumor, 20 mg of frozen tissues were ground to a fine powder by metal beads in a cryogenic homogenizer and extracted with acetonitrile-methanol-$KH_2PO_4$ (*50:30:20*) buffer. Measurements of metabolites (Zallot *et al*, 2013; Palmieri *et al*, 2014, 2017b) and glufosinate (Wang *et al*, 2008) were obtained with an Acquity UPLC system interfaced with a Quattro Premier mass spectrometer (Waters).

### Tumor grafts

For animal studies, a "single-blind" design was used. 1 × 10^6 Lewis lung carcinoma (CD90.1^+) cells and YUMM1.7 (CD90.1^+) melanoma cells were injected subcutaneously and intradermally, respectively, into a shaved rear flank of 8-week-old C57BL/6 mice, whereas 1 × 10^6 4T1 (GFP^+) mammary carcinoma cells, suspended in 200 μl PBS, were orthotopically inoculated to the mammary fat pad of 8-week-old BALB/c mice. Following tumor growth, mice randomization was performed to reach same tumor average per cage and normalize the pharmacological effects on metastasis. Tumor volumes were measured three times a week with a caliper and calculated using the formula: $V = \pi \times (d2 \times D)/6$, where d is the minor tumor axis and D is the major tumor axis. Glufosinate solution was prepared in sterile water for injection and administered directly into the stomach of mice twice a day via oral gavage. Tumor-bearing mice received glufosinate until end stage, after tumors reached volume of 80–100 mm^3. At the end stage, tumor weight was measured and tumors, lungs, and other organs were collected for histological examination. In adjuvant therapeutic regimen setting, LLC-bearing mice received glufosinate treatment 14 days after tumors reached volume of 400 mm^3 and were resected. The post-surgical mice were sacrificed at end stage, and the lung metastases were analyzed.

### Extravasation assay

GFP-expressing 4T1 subclones (1 × 10^6 cells) were injected intravenously into 9-week-old BALB/c female mice preconditioned with glufosinate for 2 days. After 2 days of treatment, to determine the overall lung metastatic breast cancer cell burden, genomic DNA was purified from the contralateral lung, subjected to quantitative real-time PCR (qPCR) using specific primers for GFP (Forward: CATGGTCCTGCTGGAGTTCGTG; Reverse: CGTCGCCGTCCAGCTC GACCAG; BD Biosciences) and the results were normalized to the vehicle group.

### Neoadjuvant therapeutic regimen, intravasation, and survival assay

1 × 10^6 GFP-labeled 4T1 cells were injected in the mammary fat pad of 9-week-old BALB/c female mice. After 14 days of treatment, at the time of tumor resection, blood was collected (300 μl) by retro-orbital bleeding and subjected to hemolysis. Genomic DNA was purified and subjected to quantitative real-time PCR (qPCR) using specific primers for GFP (as above), and the results were normalized to the vehicle group. Survival studies were performed until the natural death of the post-surgical mice.

## Metastasis analysis

Lung metastasis nodules were contrasted after intratracheal injection of 15% India ink solution and assessed under a stereomicroscope. Where indicated, the lungs were embedded in paraffin, sectioned, and stained with Hematoxylin and eosin (H&E) to check metastatic lesion area. The lung metastatic lesions originated from primary melanoma were checked by DAB-H stain for CD90.1 positivity. The area of lung metastasis was measured using ImageJ software (Bethesda, MD, USA).

## FACS analysis and flow sorting of tissue- and tumor-associated macrophages

LLC or YUMM tumor-bearing mice were sacrificed by cervical dislocation, and tumors were harvested. Tumors or other organs were minced in RPMI medium containing 0.1% collagenase type I and 0.2% dispase type I and incubated in the same solution for 30 min at 37°C. Samples from spleen were mechanically dissociated. The digested or dissociated tissues were filtered using a 70 μm pore sized mesh, and cells were centrifuged 5 min at 1,200 *g*. Red blood cell lysis was performed by using Hybri-MaxTM (Sigma-Aldrich). For flow sorting, the myeloid cell population in the tumor single cell suspension, and when appropriate in flushed splenocytes, was enriched with CD11b-conjugated magnetic beads (MACS, Miltenyi Biotec) and separated through magnetic columns (MACS, Miltenyi Biotec). Cells were resuspended in FACS buffer (PBS containing 2% FBS and 2 mM EDTA) and incubated for 15 min with Mouse BD Fc Block™ purified anti-mouse CD16/CD32 mAb (BD Biosciences, 553142) and stained with the following antibodies for 30 min at 4°C in dark: anti-CD4 (BioLegend,100540), anti-CD8 (eBioscience, 53-0081-82), anti-MHCII (eBioscience, 46-5321-82), anti-CD206 (BioLegend, 321120), anti-CD11c (eBioscience, 17-0114-81), anti-CD69 (BioLegend,104522), anti-41BB (Thermo Fisher Scientific, 25-1371-80), anti-CD45 (BioLegend, 103108), anti-F4/80 (BioLegend, 123128), anti-TCRbeta (Thermo Fisher Scientific, 17-5796-82), anti-CD25 (eBioscience, 25-0251-82). Cells were subsequently washed and resuspended in cold FACS buffer before FACS analysis or flow sorting by a FACS Verse or FACS Aria (BD Biosciences), respectively. Suitable negative isotype controls were used to rule out the background fluorescence for the used fluorochromes (FITC, PE, PE-Cy5, PE-Cy7, and APC). To ensure that analyses were made only of viable cells and not debris, all events labeled with 7AAD were excluded. Percentage of each positive population and mean fluorescence intensity (MFI) were determined using quadrant statistics. More details about reagents and dilutions are reported in Appendix Table S1.

## Immunostainings

For serial sections cut at 7 μm thickness, tissue samples were fixed in 2% PFA overnight at 4°C, dehydrated, and embedded in paraffin. Paraffin slides were first rehydrated to further proceed with antigen retrieval in citrate solution (DAKO). The sections were blocked with the appropriate serum (DAKO) and incubated overnight with the following antibody: rat anti-CD31 (BD, 557355) 1:200, rat anti-F4/80 (Bio-Rad, MCA497G) 1:100, rabbit anti-hypoxyprobe (NPS, HP3-100KIT) 1:100, rat anti-Thy1.1 (Abcam, ab85352) 1:200, rat anti-

actin α-smooth muscle (Sigma-Aldrich, C6198-.2ML) 1:500, goat anti-CD206 (R&D Systems, AF2535) 1:100, rabbit anti-CA9 (Novus Biologicals, NB100-417) 1:100, rabbit anti-FITC (Bio-Rad, 4510-7604) 1:100, hamster anti-CD11c (eBioscience, 14-0114-81) 1:200. Appropriate secondary antibodies were used: Alexa 488 (Invitrogen, A11001)-, 647 (Invitrogen, A21447)-, or 568 (Invitrogen, A11057)-conjugated secondary antibodies 1:200, biotin-labeled antibodies (bio-connect: 711-065-152 and 705-065-003) 1:300 and, when necessary, TCA fluoricine or TSA Plus Cyanine 3 System amplification (Perkin Elmer, Life Sciences) were performed according to the manufacturer's instructions. Whenever sections were stained in fluorescence or IHC, ProLong Gold mounting medium with or without DAPI (Invitrogen) and DPX mounting were used, respectively. Microscopic analysis was done with an Olympus BX41 microscope and cellSens imaging software. More details about reagents and dilutions are reported in Appendix Table S1.

## Mice blood analysis

Whole blood from the retro-orbital sinus was collected into EDTA-treated tubes and used for red and white blood cell counting by CBC disposable hemacytometer. For plasma analysis, cells were removed from plasma by centrifugation for 15 min at 2,000 *g* using a refrigerated centrifuge. The resulting supernatant designated plasma was used for glutamine quantification by mass spectrometry. For serum analysis, whole blood as above was collected into untreated tubes. After collection, whole blood was left undisturbed 30 min at room temperature to allow the clot. The clot was removed by centrifuging at 2,000 *g* for 10 min in a refrigerated centrifuge. The resulting supernatant designated serum was used for hepatic enzymes levels quantification.

## *Ex vivo* culture of CTCs

A cardiac puncture was executed to harvest arterial blood that was kept on ice in lithium heparin Microtainer Tubes (BD Biosciences). 100 μl of blood was seeded in a 10-cm culture dish in 10 ml DMEM plus 10% FCS and 1% penicillin-streptomycin. After 14 days of culture, cancer cell colonies were evaluated under a fluorescence microscope. The total number of colonies was quantified manually in 2 or more petri dishes for each animal on an Olympus IX71 microscope (La Porta *et al*, 2018).

## Tumor homogenates

For the preparation of whole tumor lysates, 20 mg of frozen mouse tumors was ground to a fine powder by sand beads and homogenized in 500 μl of extraction buffer (10 mM Tris, pH 7.6, 5 mM EDTA, 50 mM NaCl, and 1% Triton X-100) with protease inhibitor cocktail (set I, Calbiochem) and phosphatase inhibitor cocktail (set I and II, Sigma). The homogenates were centrifuged at 12,000 *g* for 20 min at 4°C, and the supernatants were used for Western blotting analysis.

## Tumor interstitial fluid

Tumor interstitial fluids were collected by centrifugation method as reported (Wiig *et al*, 2003), and the secreted IFN-γ was quantified by ELISA (Abcam, ab46081).

## Cancer cell invasion assay

The upper side of a Transwell chamber with a 8.0 μm-porous poly-carbonate membrane filter (Costar®) was coated with 5 μg Matrigel (Matrix Growth Factor Reduced from Corning®). M2 macrophages pretreated or not with MSO/glufosinate were washed, detached in cold PBS, and then added ($5 \times 10^4$/well) together with calcein-labeled A549 or LLC cells ($1 \times 10^5$/well) to the upper chamber in RPMI medium supplemented with 2% FBS. RPMI medium supplemented with 10% FBS was added to the lower well to create a gradient of serum as chemoattractant stimulus. The migration of A549 or LLC cells was assayed after 24 h of incubation. At this time point, the membranes were removed and stained with DAPI, and the number of cells that had invaded the lower chamber was counted in three randomly selected fields under a fluorescence microscopy.

## Cancer cell proliferation

Murine BMDMs pretreated or not with glufosinate and polarized into M2 macrophages were washed in PBS and added ($5 \times 10^4$/well) in glutamine-free medium to the upper side of a Transwell chamber with a 0.4 μm-porous polycarbonate membrane filter (Costar®). Murine LLC cells ($1 \times 10^5$/well) were added to the lower chamber in glutamine-free medium, and after 72 h of incubation, the cell number was measured spectrophotometrically by using cell counting kit-8 (Dojindo) at 450 nm.

## Effects of glufosinate on cancer cells proliferation, migration, and viability

Murine cancer cells ($1 \times 10^5$/well) were treated directly with glufosinate in normal and glutamine-free medium and monitored for 72 h by using the CellTiter 96® Non-Radioactive Cell Proliferation Assay (Promega) as described in (Menga et al, 2017; Palmieri et al, 2017b). The migration of glufosinate-treated cancer cells was evaluated in normal and glutamine-free medium as described before, but without macrophages.

## Endothelial cell capillary formation

$2 \times 10^5$ M2 polarized, human macrophages, pretreated or not with MSO/glufosinate, were washed in PBS and embedded in Matrigel (BD biosciences). After 4 h, $1 \times 10^4$ Human Umbilical Vein Endothelial Cells (HUVEC) were added for an additional 4 h to the Matrigel. HUVEC capillary network was measured by counting the number and length of branches using ImageJ software.

## Blood vessel perfusion and hypoxia assessment

Tumor-bearing mice were injected intravenously with 0.05 mg fluorescein isothiocyanate (FITC)-conjugated lectin (Lycopersicon esculentum; Vector Laboratories) and perfused (lectin$^+$) tumor vessels were counted on tumor sections. Tumor-bearing mice were injected i.p. with 60 mg/kg pimonidazole hydrochloride and tumor hypoxic (pimonidazole$^+$) areas were evaluated on paraffin sections immunostained with the Hypoxyprobe-1-Mab1 (Hypoxyprobe kit, Chemicon) following the manufacturer's instructions.

## T-cell purification and expansion

CD8$^+$ T cells were isolated by negative selection using CD8$^+$ T Cell Isolation Kit (Miltenyi). Fresh cells after isolation were then washed with PBS and resuspended in RPMI 1640 medium supplemented with 10% FCS, 10 mM HEPES, 2 mM glutamine, and non-essential amino acids. Purity of cells was checked by flow cytometric analysis and cell population with > 90% purity was taken. These cells were activated and expanded. To address the immunopromoting function of macrophages, cultured primary IL10-polarized macrophages, with or without a previous 2-h incubation with glufosinate/MSO and/or rapamycin, were seeded in 96-well plates ($2 \times 10^5$/ml) and cocultured with CD8$^+$ T cells in stimulator–responder ratios of 1:2. Macrophage stimulants and treatments were washed away extensively before adding T-cell suspensions in a final volume of 200 μl. Cells were cocultured for 5 days in the presence of anti-CD3/CD28 Dynabeads (Life Technologies) and IL-2 25 U/ml for activation (Palmieri et al, 2017b). As controls, $1 \times 10^5$ CD8$^+$ T cells/well were also plated without macrophages and left alone throughout the entire procedure. T cells were collected after 5 d in culture, and after incubation with 1 μCi/well tritiated thymidine, their proliferation was evaluated by reading radioactivity as cpm (counts per minute). Proliferation measured in T cells without macrophages was set at 100%. Where indicated, cocultures of macrophages and CD8$^+$ T cells were kept in normoxia or hypoxia (1% oxygen) throughout the entire procedure.

## T-cell migration assay

Migration of CD8$^+$ T cells was evaluated by using a Transwell Permeable system with 5-μm polycarbonate membrane (Costar). To determine cell migration in response to soluble factors secreted by macrophages, the last were precultured in the lower chamber for 7 days in RPMI 10% FBS and then activated for 8 h with LPS/IFNγ or IL10 with or without MSO/glufosinate. Then, the medium was changed, and after 18 h, autologous CD8$^+$ T cells at 48 h culture after isolation in the presence of anti CD3 (1 μg/ml) and anti CD28 (1 μg/ml) were placed in the upper chamber ($2 \times 10^5$ cells in 70 μl of medium with 10% FBS). After a 3 h incubation of macrophages with CD8$^+$ T cells, the amount of migrated T cells was evaluated by microscopy (Finisguerra et al, 2015).

## Moto-neuron toxicity assays

To evaluate the mouse forearm grip strength, measured in Newtons (N), a grip strength meter (Columbus Instruments, OH, USA) was employed. Mice held by the tail were allowed to grip the trapeze with their front paws and then pulled with their body parallel to the floor. Each animal was evaluated 15 times in sets of 3 with resting intervals. Neuromuscular strength was assessed in vehicle and glufosinate-treated mice, before the treatment (baseline), after 1 day (Acute Toxicity), and after 15 days (Chronic Toxicity). After excluding the highest and lowest values, the remaining readings for each mouse were averaged.(Fujii et al, 2019). In a different setting, a special grid allowing the mouse to grasp it with all four limbs was used. In this case, all-animal strength is measured. Furthermore, mice were trained daily for 3 consecutive days on the rotarod apparatus (Rotarod apparatus, May Commat RR0711, Ankara, Turkey;

rod diameter: 2 cm) at a constant speed of 18 rotations per minute (rpm) (Salat *et al*, 2015). After 24 h training, the inability to remain on the rotating rod for 300 s (indication of motor impairment) was measured in vehicle and glufosinate-treated mice, before the treatment (baseline), after 1 day (Acute Toxicity), and after 15 days (Chronic Toxicity). Then, the von Frey test was performed to evaluate a painful response to an harmless stimulus (tactile allodynia) by using an electronic von Frey unit (Bioseb, Montpellier, France) supplied with a single flexible filament applying increasing force (from 0 to 10 g) against the plantar surface of the hind paw of the mouse. Since the stimulus was turned off by the nocifensive paw withdrawal response, the instrument recorded the mechanical pressure that evoked the response. The animals were allowed to acquaint for 1 h in test compartments with a wire mesh bottom. After evaluating the baseline values, treatment with glufosinate or vehicle took place. After 1 day (acute toxicity) or 15 days (chronic toxicity), their reactivity was measured several times for each animal and the results were averaged (Salat *et al*, 2015).

### Macrophage depletion

Mice were macrophage-depleted by daily i.p. injection of 20 mg/kg mouse anti-CSF1R antibodies (clone AFS98; BioXcell) or isotype IgG control (Sigma-Aldrich). Mice were pretreated for 3 days with anti-CSF1R before LLC tumor cell injection and then injected 3 times a week until sacrifice. At the end stage, 300 μl of blood was collected by retro-orbital bleeding and subjected to hemolysis. Total RNA was isolated, treated with DNase (Qiagen), and assessed for purity by spectroscopy. A Bio-Rad iScript reverse transcription kit was used to make cDNA from 150 ng RNA. TaqMan probe and primers for CD90.1 were purchased from IDT as follows: Probe 5′-/56-FAM/AGCTTCGCG/ZEN/TCAGC/3IABkFQ/-3′, Fwd 5′-GGATGAGGGCGACTACTT-3′ and Rv 5′-ACTTGACCAGCTTGTCTCTATAC-3′. Analysis was carried out with a QuantStudio real-time PCR system (Thermo Fisher). All transcript levels were normalized against the β-actin expression level and compared to vehicle group.

### Statistics

Data entry and all analyses were performed in a blinded fashion. Statistical analysis was performed using GraphPad Prism software. Data were tested for normality using the D'Agostino–Pearson omnibus test (for $n > 8$) or the Kolmogorov–Smirnov test (for $n \leq 8$) and variation within each experimental group was assessed. Statistical significance was calculated by two-tailed unpaired *t*-test or ANOVA test with Tukey *post hoc* test. Sample sizes for all experiments were chosen based on previous experiences. Independent experiments were pooled and analyzed together whenever possible as detailed in figure legends. Results, represented as means ± SEM, were considered statistically significant as follows: *$P < 0.05$, **$P < 0.01$, ***$P < 0.001$, ****$P < 0.0001$. Exact $P$ values and statistical tests are reported for each experiment in Appendix Table S2.

### Study approval

Housing and all experimental animal procedures were approved by the Institutional Animal Care and Research Advisory Committee of the KU Leuven. For studies on macrophages from PBMCs, buffy coats were obtained from the local Blood Bank under an Institutional approved agreement (B322201215873 for Leuven and 0070295 for Bari). Additionally, experiments were conformed to the principles set out in the WMA Declaration of Helsinki and the Department of Health and Human Services Belmont Report.

**Expanded View** for this article is available online.

### Acknowledgements

MM is supported by an ERC Consolidator Grant (ImmunoFit, #773208), FWO (G0D1717N and G066515N), and Stichting Tegen Kanker (2014-197). AC is supported by grant from the Italian Ministry of Economic Development (MISE) (F/200076/01-02/X45). PEP is supported by AIRC (MFAG 21564). AM is supported by AIRC (MFAG 21564), EMBO and FEBS Fellowship grants. CR-D is supported by FWO (1108919N).

### Author contributions

AC and MM designed research; AM, MS, ST, CR-D, EMP, ME, UA, MADN, CLP, RG, MF, and PEP performed experiments; AM and ST analyzed data; AC and MM wrote the manuscript.

### Conflict of interest

The authors declare that they have no conflict of interest.

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
