## [Review Process File · EMBO Molecular Medicine]

Glufosinate constrains synchronous and metachronous metastasis by promoting anti-tumor macrophages

Alessio Menga, marina serra, simona todisco, Carla Domingo, Umami Ammarah, Manuel Ehling, Erika Palmieri, Maria Antonietta Di Noia, Rosanna gissi, Maria Favia, Ciro Pierri, Paolo Porporato, Alessandra Castegna, and Massimiliano Mazzone

DOI: [10.15252/emmm.201911210](https://doi.org/10.15252/emmm.201911210)

Corresponding authors: Alessandra Castegna (alessandra.castegna@uniba.it), Massimiliano Mazzone (massimiliano.mazzone@kuleuven.vib.be)

Review Timeline:

Submission Date:	4th Sep 19
Editorial Decision:	25th Sep 19
Revision Received:	25th Jun 20
Editorial Decision:	16th Jul 20
Revision Received:	31st Jul 20
Accepted:	1st Aug 20

Editor: Celine Carret

Transaction Report:

25th Sep 2019

Dear Prof. Mazzone,

Thank you for the submission of your manuscript to EMBO Molecular Medicine. We have now heard back from the two referees whom we asked to evaluate your manuscript.

You will see that both referees find the study of interest, however they both have overlapping concerns, including a somehow limited overall advance due to your previous publication. Should you be able to provide more mechanism along the below lines, we would consider a major revision of this work for further consideration.

- 1) the study needs to get some MoA/causality on how the drug affects metastasis
- 2) need to show the effect of glufosinate is through GS inhibition of macrophages in vivo
- 3) why are your results not in agreement with previous publications regarding GS inhibition in CAFs
- 4) effects on tumour cells to be investigated

In addition, there are a number of additional/supportive experiments recommended to make the paper stronger and more conclusive.

We would therefore welcome the submission of a revised version within three months for further consideration and would like to encourage you to address all the criticisms raised as suggested to improve conclusiveness and clarity. Please note that EMBO Molecular Medicine strongly supports a single round of revision and that, as acceptance or rejection of the manuscript will depend on another round of review, your responses should be as complete as possible.

I look forward to receiving your revised manuscript.

Yours sincerely,

Celine Carret

Celine Carret, PhD
Senior Editor
EMBO Molecular Medicine

*** Instructions to submit your revised manuscript ***

**** PLEASE NOTE **** As part of the EMBO Publications transparent editorial process initiative (see our Editorial at <https://www.embopress.org/doi/pdf/10.1002/emmm.201000094>), EMBO Molecular Medicine will publish online a Review Process File to accompany accepted manuscripts.

To submit your manuscript, please follow this link:

Link Not Available

- 1) a .doc formatted version of the manuscript text (including Figure legends and tables). Please make sure that the changes are highlighted to be clearly visible to referees and editors alike.
- 2) separate figure files*
- 3) supplemental information as Expanded View and/or Appendix. Please carefully check the authors guidelines for formatting Expanded view and Appendix figures and tables at <https://www.embopress.org/page/journal/17574684/authorguide#expandedview>
- 4) a letter INCLUDING the reviewers' reports and your detailed responses to their comments (as Word file)

Also, and to save some time should your paper be accepted, please read below for additional information regarding some features of our research articles:

- 5) The paper explained: EMBO Molecular Medicine articles are accompanied by a summary of the articles to emphasize the major findings in the paper and their medical implications for the non-specialist reader. Please provide a draft summary of your article highlighting
 - the medical issue you are addressing,
 - the results obtained and
 - their clinical impact.

6) For more information: There is space at the end of each article to list relevant web links for further consultation by our readers. Could you identify some relevant ones and provide such information as well? Some examples are patient associations, relevant databases, OMIM/proteins/genes links, author's websites, etc...

7) Author contributions: the contribution of every author must be detailed in a separate section (before the acknowledgments).

8) EMBO Molecular Medicine now requires a complete author checklist (<https://www.embopress.org/page/journal/17574684/authorguide>) to be submitted with all revised manuscripts. Please use the checklist as a guideline for the sort of information we need WITHIN the manuscript as well as in the checklist. This is particularly important for animal reporting, antibody dilutions (missing) and exact p-values and n that should be indicated instead of a range.

9) Every published paper now includes a 'Synopsis' to further enhance discoverability. Synopses are displayed on the journal webpage and are freely accessible to all readers. They include a short stand first (maximum of 300 characters, including space) as well as 2-5 one sentence bullet points that summarise the paper. Please write the bullet points to summarise the key NEW findings. They should be designed to be complementary to the abstract - i.e. not repeat the same text. We encourage inclusion of key acronyms and quantitative information (maximum of 30 words / bullet point). Please use the passive voice. Please attach these in a separate file or send them by email, we will incorporate them accordingly.

You are also welcome to suggest a striking image or visual abstract to illustrate your article. If you do please provide a jpeg file 550 px-wide x 400-px high.

10) A Conflict of Interest statement should be provided in the main text

11) Please note that we now mandate that all corresponding authors list an ORCID digital identifier. This takes <90 seconds to complete. We encourage all authors to supply an ORCID identifier, which will be linked to their name for unambiguous name identification.

Currently, our records indicate that there is no ORCID associated with your account.

Please click the link below to provide an ORCID:

Link Not Available

12) The system will prompt you to fill in your funding and payment information. This will allow Wiley to send you a quote for the article processing charge (APC) in case of acceptance. This quote takes into account any reduction or fee waivers that you may be eligible for. Authors do not need to pay any fees before their manuscript is accepted and transferred to our publisher.

Photos 400-800 DPI

*Additional important information regarding figures and illustrations can be found at <http://bit.ly/EMBOPressFigurePreparationGuideline>

***** Reviewer's comments *****

Referee #1 (Comments on Novelty/Model System for Author):

Technical quality - reason for this score is that some of the markers used from which to draw conclusions are not the most direct, and could also result in alternative conclusions
Novelty - this manuscript is a follow-up to a previous manuscript published in Cell Metabolism. There are areas of novelty within the manuscript
Impact - medium as the compound used could feasibly be translated into initial studies in man

Referee #1 (Remarks for Author):

The manuscript from Menga et al. details a study in which the authors show that glufosinate is able to inhibit glutamine synthetase (GS), and through this alter the phenotype of macrophages from an M1-like to an M2-like polarization. They also show data that reports on the effects of glufosinate on tumor growth and metastasis using a significant number of mouse models. The manuscript directly follows on from their previous publication (Palmieri et al. Cell Reports [2017]), describing the effect of inhibiting GS using methionine sulfoximine (MSO) on macrophages - which indeed is also a re-polarization from M1-like to M2-like. The manuscript therefore has two significant areas of novelty - that glufosinate is a novel inhibitor of human GS (as the authors point out, it is already understood to inhibit GS in other organisms), and that glufosinate inhibits tumor metastasis. The significant weakness is that they have not shown that the effect of glufosinate on tumor phenotype is through its specific inhibition of GS in macrophages, in the tumor, or in the rest of the body. As they have previously shown that conditional knockout of GS in macrophages prevents metastasis (Palmieri et al. Cell Reports [2017]), this would be the presumed result. However, as we do not know whether glufosinate has alternative non-GS targets, it cannot be concluded that its effect on tumor metastasis is through macrophage-specific inhibition of GS. Indeed, as a glutamate-like molecule, it could inhibit a number of different metabolic enzymes.

The major issues are therefore:

1. Does glufosinate only inhibit GS in macrophages? Are there other metabolic targets that may also play a role?
2. GS has previously been suggested to support glutamine synthesis in cancer-associated fibroblast, and more importantly, support tumor growth (Yang et al. Cell Metabolism 2016). Inhibition of this enzyme was shown to lead to tumor regression. This is clearly in contrast to the results shown in this manuscript. Given that glufosinate is unlikely to affect GS activity only in the macrophage population (see comment in first paragraph above), could the authors address the apparent discrepancy in these results?

3. The authors have not included the data that was obtained in assessing the recombinant GS - this would be useful to have in the manuscript.
4. The authors present mRNA data regarding control of cytokine expression. However, as the authors will appreciate, it is often more relevant to know the protein expression, especially with molecules such as cytokines and chemokines. Could the authors back-up some of their data showing a phenotypic switch with quantitation of expression and/or secretion of the molecules assessed, where suitable.
5. The authors use the reference Jain et al. 2005 (page 10, line 8) to support their supposition that SMA is a marker of vessel normalization. This is not mentioned in this review article, and a more appropriate reference included instead.
6. On page 11, para 1, the conclusions are somewhat over-reaching and not supported by the data. In order to suggest how the metastases may be affected, the authors would need the data - whether intravasation, extravasation or proliferation at the distant site. Their data also do not support shrinkage of the metastatic site - for this, they would need paired observations of the metastases before and after glufosinate treatment. Indeed, the data shown in Figure 4 could also be consistent with increased metastases but decreased growth, leading to high numbers of micrometastases.
7. Use of pimonidazole as a means of demarcating hypoxic areas is well-defined, but use of MHCII expression as a surrogate is not reasonable, given that local changes in inflammatory response can also alter MHCII expression - this would be expected to alter in the setting of glufosinate treatment, as the authors' data show. A more appropriate marker of hypoxia is therefore preferred.
8. In Figure 2F, some results are somewhat confusing. The +LPS/IFN treatment appears to have no effect on T cells compared to control, contrary to what is suggested in the text. Additionally, could the authors clarify where CXCL10 is present and absent.
9. The use of different concentrations of 10 and 20 mg/kg glufosinate is inconsistent; could the authors provide a rationale for different doses being used in the in vivo experiments or present data where the same dose is used throughout - compare the use of control and 2 doses in Figure 3, control and the 20 mg/kg dose in Figures 4 and 5, while control and 10 mg/kg in Figure 6.
10. In Figure 3I the authors show pimo staining, marking hypoxic cells, and suggest a change in vascular perfusion. However, as the authors are aware, hypoxia can arise as a result of multiple factors. A more robust measure of perfusion is therefore required to support their statement, such as tomato lectin injected before cull.
11. The authors have previously shown that GS inhibition stabilizes HIF, and in this manuscript, inhibit the response of macrophages using acriflavine. They have not included data here that they observe HIF1a stabilization after glufosinate treatment, which would be useful. Assuming this is the case, could they explain why chronic dosing of mice with glufosinate does not elicit an increase in RBC number through an EPO effect (Figure 7b)?

Minor points:

1. The definition of the IL-10 treated macrophages as M2 is not ideal as this state is most often obtained through treatment with IL4 +/- IL13. The IL-10 treated macrophages are clearly still anti-inflammatory, but would be most appropriately referred to as M2-like
2. Treatment schedule for Figure 4 is unclear - when was treatment started and for how long?
3. Use of NOS2 as a marker of macrophage polarization is well-described for mouse macrophages, but not as well accepted in human macrophages.
4. In Figure 3L T cells are presented as '% of alive', while in Figure 5R and S, '% of CD45'. Could the authors make the reporting consistent.
5. Treatment schedule behind the data in Figure 7 is unclear.

Referee #2 (Comments on Novelty/Model System for Author):

The model systems employed are adequate to evaluate the questions posed and they provide the rationale for future studies.

Referee #2 (Remarks for Author):

EMBO Mol. Med.

Title: Glufosinate promotes anti-tumor TAMs and constrains synchronous and metachronous metastasis

Summary

Menga et al investigate the contribution of glutamine synthetase (GS) in macrophages to tumor progression. This is an extension of a 2017 study from this group, which showed that pharmacologic (MSO) and genetic inhibition of GS reprograms macrophages to a M1-like phenotype and reduces metastasis in a mouse model (LLC) cancer. In this report, they introduce a new inhibitor of GS, glufosinate ammonium (glufosinate), which has been developed/evaluated/tested as an inhibitor of plant GS. The authors validate that glufosinate inhibits mammalian (human) GS and compare glufosinate to MSO showing that the glufosinate has properties and an inhibitory profile that suggest it is an effective (and better) inhibitor of mammalian GS. In vitro studies with cultured human and mouse macrophages show that GS inhibition with glufosinate results in metabolic reprogramming and a shift in macrophage phenotype towards an M1-like state. Using co-culture experiments that authors show that glufosinate blunts the effect of M2-like (IL-10 treated) macrophages on tumor cell migration and endothelial cell tube formation. The authors also demonstrate that glufosinate accumulates in tumors after in vivo administration and test the efficacy of glufosinate in multiple tumor models that demonstrate glufosinate reduces metastatic burden but has no effect on primary tumor growth. Treatment with glufosinate reduces the number of CD206+ macrophages and MVD in primary tumors but increases pericyte-associated blood vessels and decreases hypoxia - characteristics associated with vascular normalization. Modest but statistically significant changes in T cells including increased CD8 and increased CD69+ CD8 cells are shown in 2 of the models (LLC and YUMM). Finally they provide strong evidence that glufosinate is well-tolerated by mice.

Reprogramming of macrophages within the tumor microenvironment is an attractive strategy that has clinical relevance. Glufosinate ammonium appears to be an effective and safe inhibitor of GS that induces a robust switch in macrophage phenotype. Overall this is an interesting study with strong data in many respects. However, there are a few challenges that limit impact and should be addressed before further consideration. Addressing these concerns is important especially since the concept of targeting GS in macrophages has already been suggested and shown to be useful.

General Comments

1. General comment regarding clarity. In general the figure legends do not provide enough information to interpret the data displayed. This is especially noted in the figures that display data from in vivo experiments. For instance it is not clear when therapy was initiated. Additionally, two types of metastatic experiments were performed but the methods of these experiments are not described even briefly in the legends. This makes the reader/reviewer work much harder than normal to dig through the methods to determine how each experiment was performed and compare results between figures. I strongly suggest adding detail to the legends.

2. An overarching concern is that the effect of GS inhibition on tumor cell metabolism is not discussed and data on how glufosinate affects tumor cells directly is not provided. It is clear the focus on the study and this group is how the drug influences the immune microenvironment and in particular macrophages; however, in vivo the drug will be presumably inhibit GS in all cells. Given the robust anti-metastatic effects it is important to show how glufosinate effects tumor cell phenotype.

3. I anticipate this is a query that has arisen previously but the authors should comment on the apparent discrepancy between their proposed effect on HIF1 α and its importance to the macrophage phenotypic change induced by glufosinate and prior studies that have demonstrated that reducing or blocking HIF expression in macrophages results in reduced metastatic burden (see PMID 20841473 and others).

4. The anti-metastatic effect of glufosinate is clear, interesting and definitely worth pursuing. Yet how the drug actually inhibits metastasis is not clear. The authors allude to an immune effect but do not provide data demonstrating T cell-mediated killing. The authors suggest the effect is mediated by the change in macrophage phenotype and macrophages have been implicated at helping tumor cell during the metastatic cascade; yet there is no data on whether the anti-metastatic effect is due to reduced intravasation or reduced seeding. This should be addressed at least partially in the revision.

Specific comments on figures

5. Analysis at the protein level of at least some of the cytokines displayed in Fig 1-2 is essential.

6. The effect of glufosinate alone on macrophages, tumor cells and endothelial cells should be shown in appropriate panels in Fig 1 and 2.

7. Figure 2F is not convincing and is confusing since the controls (e.g., LPS/IFN γ and IL10) do not significantly alter T cell migration. If CXCL9/10 expression by Gluf treated macrophages is inducing T cell migration does neutralization of CXCL9/10 in that conditioned media reduce T cell migration?

8. It will be useful to evaluate additional M1- (iNOS) and M2 (Arg1) macrophage markers in the in vivo studies. For example, in Figure 3F, it is not convincing to specify M1 or M2-like TAMs based on CD206 expression only.

9. Survival data from any of the mouse models would greatly increase the impact of the paper.

10. Metastatic burden at the start of therapy in figure 4 is required to suggest that treatment reduces existing metastatic lesions. This could be done by harvesting multiple animals at the initiation of therapy.

11. In figure 5, evaluation of total CD11c⁺ F480⁺ cells and CD206⁺F480 cells irrespective of MHC class II expression is important. Additional markers such as iNos and arginase should also be evaluated.

12. Evaluation of CD69 is reasonable but not enough to demonstrate anti-tumor T cells. Additional T cell characterization either via flow, gene expression, or functional analysis is required.

13. Analysis of immune landscape in the metastatic site will be useful to support metastasis data in the mouse models. This could be done with existing sections by IHC.

14. The histology in Figure 6 is difficult to interpret and the images provided do not look like H&E images. Is this due the fact that india ink was perfused into the lungs? Please clarify the relevant differences between the groups.

Minor comments

15. The use of 3 distinct syngenic models is impressive. Was there any rationale for choosing these three models?

16. The manuscript would benefit from careful editing for clarity.

RESPONSE TO REVIEWERS**Manuscript Number: EMM-2019-11210**

Based on the referees' comments, we revised the present manuscript by implementing significantly the result with additional experiments whose details are described below.

Referee #1 (Remarks for Author):

The manuscript from Menga et al. details a study in which the authors show that glufosinate is able to inhibit glutamine synthetase (GS), and through this alter the phenotype of macrophages from an M1-like to an M2-like polarization. They also show data that reports on the effects of glufosinate on tumor growth and metastasis using a significant number of mouse models. The manuscript directly follows on from their previous publication (Palmieri *et al*, 2017), describing the effect of inhibiting GS using methionine sulfoximine (MSO) on macrophages - which indeed is also a re-polarization from M1-like to M2-like. The manuscript therefore has two significant areas of novelty - that glufosinate is a novel inhibitor of human GS (as the authors point out, it is already understood to inhibit GS in other organisms), and that glufosinate inhibits tumor metastasis. The significant weakness is that they have not shown that the effect of glufosinate on tumor phenotype is through its specific inhibition of GS in macrophages, in the tumor, or in the rest of the body. As they have previously shown that conditional knockout of GS in macrophages prevents metastasis (Palmieri *et al*, 2017), this would be the presumed result. However, as we do not know whether glufosinate has alternative non-GS targets, it cannot be concluded that its effect on tumor metastasis is through macrophage-specific inhibition of GS. Indeed, as a glutamate-like molecule, it could inhibit a number of different metabolic enzymes.

The major issues are therefore:

1. Does glufosinate only inhibit GS in macrophages? Are there other metabolic targets that may also play a role?

We have provided significant data supporting the specificity of glufosinate toward GS. To rule out an additional off-target role of glufosinate, a sequence database screening was performed by using alternatively *H.sapiens* or *Z.mays* GS sequences, searching for highly similar paralogous sequences in mammalia and plants (see Table S1). This search revealed no significant similarity between the query GS sequences and other *mammalia* or *plant paralogous* sequences, different from GS, nor

identified structurally related proteins, beyond the available GS proteins, in the PDB until today (see Table S2).

For gaining new insights about glufosinate competitive binding mechanism to GS binding region, in presence of glutamate, the available crystallized structures were manually inspected by using 3D visualizer. Thus, it was observed that several ligands participating in the reaction catalyzed by GS were crystallized in complex at the monomer monomer interface with GS (see Figure 1 D-I), namely ADP, Phosphate ion, L- methionine-S-sulfoximine phosphate (P3S), phosphoaminiphosphonic acid-adenylate ester (ANP), glutamate, citrate and the imidazopyridine inhibitor ((4-(6-bromo-3-(butylamino)imidazol(1,2-a)pyridin-2-yl)phenoxy) acetic acid) (see Table S2). Given the structural similarity of glufosinate with glutamate ligand and P3S, after superimposition of the three molecules in the GS catalytic site, it is observed that glufosinate might bind most of residues involved in the binding of glutamate ligand (in 4hpp.pdb) and P3S (i.e. in 2qc8.pdb)(see Figure 1 D-I and Table S3).

2. GS has previously been suggested to support glutamine synthesis in cancer-associated fibroblast, and more importantly, support tumor growth (Yang *et al*, 2016). Inhibition of this enzyme was shown to lead to tumor regression. This is clearly in contrast to the results shown in this manuscript. Given that glufosinate is unlikely to affect GS activity only in the macrophage population (see comment in first paragraph above), could the authors address the apparent discrepancy in these results?

We do not understand the discrepancy of our present data with the paper authored by Yang and coworkers (Yang *et al*, 2016) because the assumptions and conclusions made by the authors stem from an experimental setting that is completely different. A “contrast” would arise if these authors would have stated that GS in CAFs is not playing a pro-tumoral role whereas the focus of the current manuscript is clearly on macrophages.

Yang *et al*. convey GS siRNA with chitosan nanoparticles, which is far from being CAF-specific (Yang *et al*, 2016). This means that also in their models, contribution of GS inhibition from other sources cannot be ruled out. Additionally, the authors did not consider that chitosan alone is known to polarize macrophages (Reichel *et al*, 2019). Finally, but most importantly, the murine (**immunological**) background and tumor model are completely different, as **nude mice** were **orthotopically** injected with the **human ovarian cancer cell line SKOV3ip**. For each of the reasons stated above, the results cannot be compared to ours; therefore, complementarity rather than discrepancy should be issued.

However, we understood that our previous experimental setting did not address specifically whether the effect on metastasis depended on macrophagic or GS inhibition in other cells, although (and importantly) we had never stated in the manuscript that other cells besides TAMs may also contribute

to the observed phenotype. To respond to this question, we treated LLC-tumor bearing mice with glufosinate alone or in combination with anti-CSF1R, in order to deplete TAMs. In this way we were able to demonstrate that the potential contribution of GS inhibition to metastasis reduction in cells different from macrophages is negligible. Besides this experimental evidence, we have indicated in the discussion that the role of GS in other cells such as endothelial cells (Eelen *et al*, 2018) and CAFs (Yang *et al*, 2016) could partly, but not mostly, contribute to the anti-tumoral / anti-metastatic effects of glufosinate.

3. The authors have not included the data that was obtained in assessing the recombinant GS - this would be useful to have in the manuscript.

We agree with the Reviewer. We have now included the immunoblotting analysis of recombinant purified GS with the anti-GS antibody (Figure 1A).

4. The authors present mRNA data regarding control of cytokine expression. However, as the authors will appreciate, it is often more relevant to know the protein expression, especially with molecules such as cytokines and chemokines. Could the authors back-up some of their data showing a phenotypic switch with quantitation of expression and/or secretion of the molecules assessed, where suitable.

In agreement with the comment from the second reviewer as well, we included the protein levels of TNF α (as M1 cytokine) and CCL18 (as M2 cytokine) (Figure S1B). Intra-tumoral interferon-gamma has been also measured (Figure 4I).

5. The authors use the reference Jain *et al*. 2005 (page 10, line 8) to support their supposition that SMA is a marker of vessel normalization. This is not mentioned in this review article, and a more appropriate reference included instead.

We included the correct reference.

6. On page 11, para 1, the conclusions are somewhat over-reaching and not supported by the data. In order to suggest how the metastases may be affected, the authors would need the data - whether intravasation, extravasation or proliferation at the distant site. Their data also do not support shrinkage of the metastatic site - for this, they would need paired observations of the metastases before and after

glufosinate treatment. Indeed, the data shown in Figure 4 could also be consistent with increased metastases but decreased growth, leading to high numbers of micrometastases.

Our data on the metachronous setting indicated that one of the mechanism of action of glufosinate is growth inhibition of established metastatic lesions. In agreement with the reviewer, we assessed how glufosinate affects other steps of the metastatic cascade:

- In order to measure the effect of glufosinate on cancer cell intravasation, we evaluated the amount of circulating cancer cells in the blood of 4T1 orthotopically injected mice at the time of tumor resection (Finisguerra *et al*, 2016; Wenes *et al*, 2016; Celus *et al*, 2017). See Figure 8I-K.
- To measure the effect of glufosinate on seeding and extravasation of circulating tumor cells at a distant metastatic site, 4T1 cells were intravenously injected to mimic metastatic progression independent of the primary tumor. Mice were sacrificed after glufosinate treatment and the lung metastatic foci were analyzed. In this way we found that glufosinate affects the cancer cell extravasation, and cancer cell survival/growth immediately after spreading. See Figure 8M-N.
- Metastatic lungs were stained and quantified for macrophage density and polarization markers. See Figure 8G-H.

7. Use of pimonidazole as a means of demarcating hypoxic areas is well-defined, but use of MHCII expression as a surrogate is not reasonable, given that local changes in inflammatory response can also alter MHCII expression - this would be expected to alter in the setting of glufosinate treatment, as the authors' data show. A more appropriate marker of hypoxia is therefore preferred.

We agree with the reviewer. We have now used MHCII, alone or in combination with other markers (i.e., CD11c and CD206), to characterise the increase in M1-like TAMs and the decrease in M2-like TAMs upon glufosinate treatment (Figure 6E-F;H-I). MHCII expression *per se* has been now used also as a readout of the antigen presenting potential of TAMs in response to glufosinate (Figure 6K-M). We have carefully re-phrased this part in the text (page 14), whereas we have used CA9 as a surrogate marker for hypoxia (Figure 7D).

8. In Figure 2F, some results are somewhat confusing. The +LPS/IFN treatment appears to have no effect on T cells compared to control, contrary to what is suggested in the text. Additionally, could the authors clarify where CXCL10 is present and absent.

We acknowledge that in the previous setting the effect on T cells of IFN γ /LPS treated macrophages was not statistically different from that of IL10 treated macrophages. In the present version of the graph we have enlarged the sample size and improved statistical analysis. We now show that IFN γ /LPS treated macrophages significantly increase T cell recruitment compared to unstimulated macrophages. The legend to Figure 2T clearly indicates that CXCL10 was used as positive control.

9. The use of different concentrations of 10 and 20 mg/kg glufosinate is inconsistent; could the authors provide a rationale for different doses being used in the in vivo experiments or present data where the same dose is used throughout - compare the use of control and 2 doses in Figure 3, control and the 20 mg/kg dose in Figures 4 and 5, while control and 10 mg/kg in Figure 6.

In the revised version of the manuscript we are explaining precisely the rationale. Once validated two doses, we used always the highest tolerated one. In C57Bl6 mice, this is 20mg/kg, whereas in BalbC mice, this is 10 mg/kg.

10. In Figure 3I the authors show pimo staining, marking hypoxic cells, and suggest a change in vascular perfusion. However, as the authors are aware, hypoxia can arise as a result of multiple factors. A more robust measure of perfusion is therefore required to support their statement, such as tomato lectin injected before cull.

This issue was addressed by systemic injection of lectin-FITC and count of FITC-positive tumor blood vessels (Figure 4D).

11. The authors have previously shown that GS inhibition stabilizes HIF, and in this manuscript, inhibit the response of macrophages using acriflavine. They have not included data here that they observe HIF1 α stabilization after glufosinate treatment, which would be useful. Assuming this is the case, could they explain why chronic dosing of mice with glufosinate does not elicit an increase in RBC number through an EPO effect (Figure 7b)?

To address this issue, we performed and included western blots for HIF-1 α in normoxic and hypoxic macrophages (Figure 2U). About the second statement, we do not agree with the Reviewer since: **i)** based on literature data the axis HIF2A-EPO is responsible for the increase of RBC whereas HIF1A protects from erythrocytosis (Franke *et al*, 2013); **ii)** many other factors besides HIF(s) play an important role in the regulation of the hematocrit such as iron homeostasis in macrophages, in liver and spleen; **iii)** EPO-producing cells are exposed to changes in oxygen tensions and it is thus possible

that, in this context, hypoxia overrules the control of HIF imposed by glufosinate. Therefore, we are not surprised that a **2 week-long** treatment with glufosinate does not affect RBC number.

Minor points:

1. The definition of the IL-10 treated macrophages as M2 is not ideal as this state is most often obtained through treatment with IL4 +- IL13. The IL-10 treated macrophages are clearly still anti-inflammatory, but would be most appropriately referred to as M2-like

We agree with the Reviewer. We are now referring to IL10-stimulated macrophages as M2-like.

2. Treatment schedule for Figure 4 is unclear - when was treatment started and for how long?

This issue was addressed by adding clear details in the legends and providing a schematic for each experimental design.

3. Use of NOS2 as a marker of macrophage polarization is well-described for mouse macrophages, but not as well accepted in human macrophages.

In the present version of the manuscript we have included additional M1 markers, such as CD80, CD86, and CXCL9 (Orecchioni *et al*, 2019) (Figure 2K-M).

4. In Figure 3L T cells are presented as '% of alive', while in Figure 5R and S, '% of CD45'. Could the authors make the reporting consistent.

This issue was addressed.

5. Treatment schedule behind the data in Figure 7 is unclear.

This issue was addressed.

References

- Celus W, Di Conza G, Oliveira AI, Ehling M, Costa BM, Wenes M & Mazzone M (2017) Loss of Caveolin-1 in Metastasis-Associated Macrophages Drives Lung Metastatic Growth through Increased Angiogenesis. *Cell Rep.* **21**: 2842–2854
- Eelen G, Dubois C, Cantelmo AR, Goveia J, Bruning U, DeRan M, Jarugumilli G, van Rijssel J, Saladino G, Comitani F, Zecchin A, Rocha S, Chen R, Huang H, Vandekerke S, Kalucka J, Lange C, Morales-Rodriguez F, Cruys B, Treps L, et al (2018) Role of glutamine synthetase in angiogenesis beyond glutamine synthesis. *Nature* **561**: 63–69
- Finisguerra V, Prenen H & Mazzone M (2016) Preclinical and clinical evaluation of MET functions in cancer cells and in the tumor stroma. *Oncogene* **35**: 5457–5467
- Franke K, Kalucka J, Mamlouk S, Singh RP, Muschter A, Weidemann A, Iyengar V, Jahn S, Wiczorek K, Geiger K, Muders M, Sykes AM, Poitz DM, Ripich T, Otto T, Bergmann S, Breier G, Baretton G, Fong GH, Greaves DR, et al (2013) HIF-1 α is a protective factor in conditional PHD2- β deficient mice suffering from severe HIF-2-induced excessive erythropoiesis. *Blood* **121**: 1436–1445
- Orecchioni M, Ghosheh Y, Pramod AB & Ley K (2019) Macrophage Polarization: Different Gene Signatures in M1(LPS+) vs. Classically and M2(LPS-) vs. Alternatively Activated Macrophages. *Front. Immunol.* **10**: 1084
- Palmieri EM, Menga A, Martín-Pérez R, Quinto A, Riera-Domingo C, De Tullio G, Hooper DC, Lamers WH, Ghesquière B, McVicar DW, Guarini A, Mazzone M & Castegna A (2017) Pharmacologic or Genetic Targeting of Glutamine Synthetase Skews Macrophages toward an M1-like Phenotype and Inhibits Tumor Metastasis. *Cell Rep.* **20**: 1654–1666
- Reichel D, Tripathi M & Perez JM (2019) Biological effects of nanoparticles on macrophage polarization in the tumor microenvironment. *Nanotheranostics* **3**: 66–88
- Wenes M, Shang M, Di Matteo M, Goveia J, Martín-Pérez R, Serneels J, Prenen H, Ghesquière B, Carmeliet P & Mazzone M (2016) Macrophage Metabolism Controls Tumor Blood Vessel Morphogenesis and Metastasis. *Cell Metab.* **24**: 701–715
- Yang L, Achreja A, Yeung TL, Mangala LS, Jiang D, Han C, Baddour J, Marini JC, Ni J, Nakahara R, Wahlig S, Chiba L, Kim SH, Morse J, Pradeep S, Nagaraja AS, Haemmerle M, Kyunghye N, Derichsweiler M, Plackemeier T, et al (2016) Targeting Stromal Glutamine Synthetase in Tumors Disrupts Tumor Microenvironment-Regulated Cancer Cell Growth. *Cell Metab.* **24**: 685–700

RESPONSE TO REVIEWERS

Manuscript Number: EMM-2019-11210

Referee #2 (Remarks for Author):

Menga et al investigate the contribution of glutamine synthetase (GS) in macrophages to tumor progression. This is an extension of a 2017 study from this group, which showed that pharmacologic (MSO) and genetic inhibition of GS reprograms macrophages to a M1-like phenotype and reduces metastasis in a mouse model (LLC) cancer. In this report, they introduce a new inhibitor of GS, glufosinate ammonium (glufosinate), which has been developed/evaluated/ tested as an inhibitor of plant GS. The authors validate that glufosinate inhibits mammalian (human) GS and compare glufosinate to MSO showing that the glufosinate has properties and an inhibitory profile that suggest it is an effective (and better) inhibitor of mammalian GS. In vitro studies with cultured human and mouse macrophages show that GS inhibition with glufosinate results in metabolic reprogramming and a shift in macrophage phenotype towards an M1-like state. Using co-culture experiments that authors show that glufosinate blunts the effect of M2-like (IL-10 treated) macrophages on tumor cell migration and endothelial cell tube formation. The authors also demonstrate that glufosinate accumulates in tumors after in vivo administration and test the efficacy of glufosinate in multiple tumor models that demonstrate glufosinate reduces metastatic burden but has no effect on primary tumor growth. Treatment with glufosinate reduces the number of CD206+ macrophages and MVD in primary tumors but increases pericyte-associated blood vessels and decreases hypoxia - characteristics associated with vascular normalization. Modest but statistically significant changes in T cells including increased CD8 and increased CD69+ CD8 cells are shown in 2 of the models (LLC and YUMM). Finally they provide strong evidence that glufosinate is well-tolerated by mice.

Reprogramming of macrophages within the tumor microenvironment is an attractive strategy that has clinical relevance. Glufosinate ammonium appears to be an effective and safe inhibitor of GS that induces a robust switch in macrophage phenotype. Overall this is an interesting study with strong data in many respects. However, there are a few challenges that limit impact and should be addressed before further

consideration. Addressing these concerns is important especially since the concept of targeting GS in macrophages has already been suggested and shown to be useful.

General Comments

1. General comment regarding clarity. In general the figure legends do not provide enough information to interpret the data displayed. This is especially noted in the figures that display data from in vivo experiments. For instance it is not clear when therapy was initiated. Additionally, two types of metastatic experiments were performed but the methods of these experiments are not described even briefly in the legends. This makes the reader/reviewer work much harder than normal to dig through the methods to determine how each experiment was performed and compare results between figures. I strongly suggest adding detail to the legends.

We agree with the Reviewer. We have now added clear details in the legends and provided a clear schematic for each experimental design.

2. An overarching concern is that the effect of GS inhibition on tumor cell metabolism is not discussed and data on how glufosinate affects tumor cells directly is not provided. It is clear the focus on the study and this group is how the drug influences the immune microenvironment and in particular macrophages; however, in vivo the drug will be presumably inhibit GS in all cells. Given the robust anti-metastatic effects it is important to show how glufosinate effects tumor cell phenotype.

In the present version of the manuscript we have included the effect of glufosinate on viability, proliferation and invasiveness of LLC, 4T1 and YUMM cells in glutamine-enriched and glutamine-deprived medium (Figure S2E-G). Second, the contribution of macrophagic GS inhibition specifically at the metastatic site and in the different steps of the metastatic cascade has been investigated in detail (Figure 8I-K;M-N). Third, the contribution of GS inhibition on the macrophage compartment vs. other compartments has been supported by carrying experiments in mice treated with glufosinate in a context where macrophages were depleted by anti-CSF1R antibodies (Figure 5D-F;S3). Finally, despite the focus of our manuscript is to underline the relevance for cancer therapy of pharmacological inhibition of GS in macrophages as a logic follow-up to our previous publication (Palmieri *et al*, 2017), we are well aware and we have never denied that the blockade of GS in other cells (i.e., cancer cells, endothelial cells and cancer-associated macrophages). This consideration has been further elaborated in the discussion, supported by accompanied literature evidence (Eelen *et al*, 2018; Yang *et al*, 2014; Tardito *et al*, 2015).

3. I anticipate this is a query that has arisen previously but the authors should comment on the apparent discrepancy between their proposed effect on HIF1 α and its importance to the macrophage phenotypic change induced by glufosinate and prior studies that have demonstrated that reducing or blocking HIF expression in macrophages results in reduced metastatic burden (see PMID 20841473 and others).

At first glance, our findings that GS-inhibited macrophages counter T cell immunosuppression and reduce metastasis by engaging HIF1 α seems in sharp contrast with the pro-tumoral, immunosuppressive role of HIF1 α in hypoxic TAMs (Doedens *et al*, 2010). However, we now solve this conundrum. We show that GS inhibition maintains mTOR active even in hypoxic conditions. This occurs through downregulation of REDD1, a known negative regulator of mTOR in hypoxia (Figure 2U-V). Active mTOR in hypoxia shields macrophages from the immunosuppressive function dictated by hypoxia forcing macrophages to display the classical functional program of normoxic pro-inflammatory M1-like macrophages, wherein mTOR is physiologically active. Concomitant activation of HIF1 α and mTOR, and in particular TORC1 (Covarrubias *et al*, 2015), is responsible for the acquisition of an immunostimulatory and antitumoral macrophage phenotype. Our results clearly demonstrate that hypoxic TAMs can engage in a **pseudo-normoxic** state, which is triggered by GS inhibition.

4. The anti-metastatic effect of glufosinate is clear, interesting and definitely worth pursuing. Yet how the drug actually inhibits metastasis is not clear. The authors allude to an immune effect but do not provide data demonstrating T cell-mediated killing. The authors suggest the effect is mediated by the change in macrophage phenotype and macrophages have been implicated at helping tumor cell during the metastatic cascade; yet there is no data on whether the anti-metastatic effect is due to reduced intravasation or reduced seeding. This should be addressed at least partially in the revision.

Our data on the metachronous setting indicated that one of the mechanisms of action of glufosinate was growth inhibition of established metastatic lesions. In agreement with the reviewer, we pursued further assessment of how glufosinate affects other steps of the metastatic cascade:

- In order to measure the effect of glufosinate on cancer cell intravasation, we evaluated the amount of circulating cancer cells in the blood of 4T1 orthotopically injected mice at the time of tumor resection (Finisguerra *et al*, 2016; Wenes *et al*, 2016a; Celus *et al*, 2017). See Figure 8I-K.

- To measure the effect of glufosinate on seeding and extravasation of circulating tumor cells at a distant metastatic site, 4T1 cells were intravenously injected to mimic metastatic progression independent of the primary tumor. Mice were sacrificed after glufosinate treatment and the lung metastatic foci were analyzed. In this way we found that glufosinate affects the cancer cell extravasation, and cancer cell survival/growth immediately after spreading. See Figure 8M-N.
- Metastatic lungs were stained and quantified for macrophage density and polarization markers. See Figure 8G-H.

Specific comments on figures

5. Analysis at the protein level of at least some of the cytokines displayed in Fig 1-2 is essential.

We included the requested analysis for TNF α (as M1 cytokine) and CCL18 (as M2 cytokine) (Figure S1B). Intra-tumoral interferon-gamma protein secretion was also measured (Figure 4I)

6. The effect of glufosinate alone on macrophages, tumor cells and endothelial cells should be shown in appropriate panels in Fig 1 and 2.

In all the experiments reported in Figure 2Q-V, the direct effect of glufosinate on cells co-cultured with macrophages is not reported because only macrophages were exposed to glufosinate. Indeed the drug-containing medium of macrophages was replaced with fresh medium (for T and cancer cells) or Matrigel (for endothelial cells) before performing co-culture experiments (see pages 27-29). The effect of glufosinate directly on cancer cells was addressed by performing *in vitro* experiments, as described above (Figure S2E-G).

Moreover, following the requests of the Reviewers, a possible non-TAM related effect of glufosinate was addressed *in vivo* via a TAM-depleting strategy, where LLC-tumor bearing mice were (pre)-treated with anti-CSF1R antibodies prior to glufosinate administration (Figure 5D-F;S3). Taking into account these results, we have extended the discussion describing possible roles of GS inhibition in other compartments that, at least in the tumor model tested, will have a minor implication when compared to the larger contribution played by macrophagic GS.

7. Figure 2F is not convincing and is confusing since the controls (e.g., LPS/IFN γ and IL10) do not significantly alter T cell migration. If CXCL9/10 expression by Gluf treated macrophages is inducing T cell migration does neutralization of CXCL9/10 in that conditioned media reduce T cell migration?

We acknowledge that in the previous setting the effect on T cells of IFN γ /LPS treated macrophages was not statistically different from that of IL10 treated-macrophages. In the present version of the graph we enlarged the sample size and improved statistical analysis. We now show that the IL10-stimulated macrophages induce a significant repression in T cell recruitment compared to IFN γ /LPS treated macrophages, that MSO reverts this phenotype and glufosinate does it at much higher extent. Although CXCL9/10 in glufosinate-treated IL10 macrophages are increased compared to IL10 macrophages, we cannot exclude that other mechanisms may be involved in this attraction. Indeed the highest effect on T cell recruitment is obtained when IL10 macrophages are treated with 10 μ M glufosinate (Figure 2T), which is not the concentration inducing the highest release of CXCL9/10 (Figure 2B-C). However, the understanding of all the mechanisms involved in T cell migration falls out of the scope of the current manuscript since the positive role of an M1-like phenotype on T cell recruitment and activation is generally accepted (Rolny *et al*, 2011).

8. It will be useful to evaluate additional M1- (iNOS) and M2 (Arg1) macrophage markers in the in vivo studies. For example, in Figure 3F, it is not convincing to specify M1 or M2-like TAMs based on CD206 expression only.

Consistent with the request below for the YUMM model, we added histological FACS-based proof of M1-like polarization by quantifying CD11c⁺ TAMs (Figure 3H). In addition, we sorted F4/80⁺ TAMs from glufosinate-treated and vehicle-treated tumors and measure CCL22 and ARG1 for M2, and CD80, CD86 for M1 polarization by qRT-PCR (Takeda *et al*, 2011) (Figure 3I-J).

9. Survival data from any of the mouse models would greatly increase the impact of the paper.

In the present version of the manuscript, we performed a survival experiment on 4T1 tumor bearing mice, following a neoadjuvant therapeutic regimen. A Kaplan-Meier plot shows that glufosinate confers significant survival advantage in treated mice compared to untreated (Figure 8L).

10. Metastatic burden at the start of therapy in figure 4 is required to suggest that treatment reduces existing metastatic lesions. This could be done by harvesting multiple animals at the initiation of therapy.

Metastatic burden at the start of therapy is now reported (Figure 5C). We treated endstage mice with overt metastatic disease (2 weeks after primary tumor resection) and evaluated the effect on metastatic progression by H&E. We found that glufosinate inhibits growth of established metastatic lesions (Figure 5A-C).

11. In figure 5, evaluation of total CD11c⁺ F480⁺ cells and CD206⁺F480 cells irrespective of MHC class II expression is important. Additional markers such as iNos and arginase should also be evaluated.

Histological proof for CD11c/CD206 TAM polarization was provided (Figure 6G;J) as in our previous papers (Bieniasz-Krzywiec *et al*, 2019; Wenes *et al*, 2016b; Casazza *et al*, 2013). Moreover, additional markers such as ARG1, CXCR4 for M2, and NOS2, CXCL9 for M1 polarization (Takeda *et al*, 2011) were measured by qRT-PCR on sorted F4/80⁺ TAMs (Figure 6N).

12. Evaluation of CD69 is reasonable but not enough to demonstrate anti-tumor T cells. Additional T cell characterization either via flow, gene expression, or functional analysis is required.

Our T cell analysis has been implemented with the evaluation of another activation marker (CD25, Figure 4H) and with the quantification of IFN γ (a cytokine produced also by activated T cells) in tumor interstitial fluid (Figure 4I).

13. Analysis of immune landscape in the metastatic site will be useful to support metastasis data in the mouse models. This could be done with existing sections by IHC.

In the present version of the manuscript, metastatic lungs were stained and quantified for macrophage numbers and polarization markers (Figure 3F; 8G-H).

14. The histology in Figure 6 is difficult to interpret and the images provided do not look like H&E images. Is this due the fact that india ink was perfused into the lungs? Please clarify the relevant differences between the groups.

These images do not look like H&E images due to the india ink perfusion into the lungs. We replaced Ink-stained lungs with regular H&E and quantifications (Figure 8F).

Minor comments

15. The use of 3 distinct syngenic models is impressive. Was there any rationale for choosing these three models?

We chose these three models as they are highly metastatic, well characterized and among the top 5 in terms of incidence in humans.

16. The manuscript would benefit from careful editing for clarity.

We carefully checked the revised text to improve clarity.

References

- Bieniasz-Krzywiec P, Martín-Pérez R, Ehling M, García-Caballero M, Pinioti S, Pretto S, Kroes R, Aldeni C, Di Matteo M, Prenen H, Tribulatti MV, Campetella O, Smeets A, Noel A, Floris G, Van Ginderachter JA & Mazzone M (2019) Podoplanin-Expressing Macrophages Promote Lymphangiogenesis and Lymphoinvasion in Breast Cancer. *Cell Metab.* **30**: 917–936
- Casazza A, Laoui D, Wenes M, Rizzolio S, Bassani N, Mambretti M, Deschoemaeker S, VanGinderachter JA, Tamagnone L, Mazzone M, Van Ginderachter JA, Tamagnone L & Mazzone M (2013) Impeding macrophage entry into hypoxic tumor areas by Sema3A/Nrp1 signaling blockade inhibits angiogenesis and restores antitumor immunity. *Cancer Cell* **24**: 695–709
- Celus W, Di Conza G, Oliveira AI, Ehling M, Costa BM, Wenes M & Mazzone M (2017) Loss of Caveolin-1 in Metastasis-Associated Macrophages Drives Lung Metastatic Growth through Increased Angiogenesis. *Cell Rep.* **21**: 2842–2854
- Covarrubias AJ, Aksoylar HI & Horng T (2015) Control of macrophage metabolism and activation by mTOR and Akt signaling. *Semin. Immunol.* **27**: 286–296
- Doedens AL, Stockmann C, Rubinstein MP, Liao D, Zhang N, DeNardo DG, Coussens LM, Karin M, Goldrath AW & Johnson RS (2010) Macrophage expression of hypoxia-inducible factor-1 α suppresses T-cell function and promotes tumor progression. *Cancer Res.* **70**: 7465–7475
- Eelen G, Dubois C, Cantelmo AR, Goveia J, Bruning U, DeRan M, Jarugumilli G, van Rijssel J, Saladino G, Comitani F, Zecchin A, Rocha S, Chen R, Huang H, Vandekeere S, Kalucka J, Lange C, Morales-Rodriguez F, Cruys B, Treps L, et al (2018) Role of glutamine synthetase in angiogenesis beyond glutamine synthesis. *Nature* **561**: 63–69
- Finisguerra V, Prenen H & Mazzone M (2016) Preclinical and clinical evaluation of MET functions in cancer cells and in the tumor stroma. *Oncogene* **35**: 5457–5467
- Palmieri EM, Menga A, Martín-Pérez R, Quinto A, Riera-Domingo C, De Tullio G, Hooper DC, Lamers WH, Ghesquière B, McVicar DW, Guarini A, Mazzone M & Castegna A (2017) Pharmacologic or Genetic Targeting of Glutamine Synthetase Skews Macrophages toward an M1-like Phenotype and Inhibits Tumor Metastasis. *Cell Rep.* **20**: 1654–1666

- Rolny C, Mazzone M, Tugues S, Laoui D, Johansson I, Coulon C, Squadrito ML, Segura I, Li X, Knevels E, Costa S, Vinckier S, Dresselaer T, Åkerud P, De Mol M, Salomäki H, Phillipson M, Wyns S, Larsson E, Buyschaert I, et al (2011) HRG inhibits tumor growth and metastasis by inducing macrophage polarization and vessel normalization through downregulation of PlGF. *Cancer Cell* **19**: 31–44
- Takeda Y, Costa S, Delamarre E, Roncal C, Leite De Oliveira R, Squadrito ML, Finisguerra V, Deschoemaeker S, Bruyère F, Wenes M, Hamm A, Serneels J, Magat J, Bhattacharyya T, Anisimov A, Jordan BF, Alitalo K, Maxwell P, Gallez B, Zhuang ZW, et al (2011) Macrophage skewing by Phd2 haploinsufficiency prevents ischaemia by inducing arteriogenesis. *Nature* **479**: 122–126
- Tardito S, Oudin A, Ahmed SU, Fack F, Keunen O, Zheng L, Miletic H, Sakariassen PØ, Weinstock A, Wagner A, Lindsay SL, Hock AK, Barnett SC, Ruppin E, Harald MØrkve S, Lund-Johansen M, Chalmers AJ, Bjerkvig R, Niclou SP & Gottlieb E (2015) Glutamine synthetase activity fuels nucleotide biosynthesis and supports growth of glutamine-restricted glioblastoma. *Nat. Cell Biol.* **17**: 1556–1568
- Wenes M, Shang M, Di Matteo M, Goveia J, Martín-Pérez R, Serneels J, Prenen H, Ghesquière B, Carmeliet P & Mazzone M (2016a) Macrophage Metabolism Controls Tumor Blood Vessel Morphogenesis and Metastasis. *Cell Metab.* **24**: 701–715
- Wenes M, Shang M, Di Matteo M, Goveia J, Martín-Pérez R, Serneels J, Prenen H, Ghesquière B, Carmeliet P, Mazzone M, Martín-Pérez R, Serneels J, Prenen H, Ghesquière B, Carmeliet P, Mazzone M, Martín-Pérez R, Serneels J, Prenen H, Ghesquière B, et al (2016b) Macrophage Metabolism Controls Tumor Blood Vessel Morphogenesis and Metastasis. *Cell Metab.* **24**: 701–715
- Yang L, Moss T, Mangala LS, Marini J, Zhao H, Wahlig S, Armaiz-Pena G, Jiang D, Achreja A, Win J, Roopaimoole R, Rodriguez-Aguayo C, Mercado-Uribe I, Lopez-Berestein G, Liu J, Tsukamoto T, Sood AK, Ram PT, Nagrath D, Armaiz-Pena G, et al (2014) Metabolic shifts toward glutamine regulate tumor growth, invasion and bioenergetics in ovarian cancer. *Mol. Syst. Biol.* **10**: 728

16th Jul 2020

Dear Prof. Castegna,

Thank you for the submission of your revised manuscript to EMBO Molecular Medicine. We have now received the enclosed reports from the referees that were asked to re-assess it. As you will see the reviewers are fully supportive and I am pleased to inform you that we will be able to accept your manuscript pending the following final editorial amendments:

1) Please provide a point-by-point letter in response to my comments and the referees' comments (as Word file).

2) Please carefully check the authors guidelines for formatting your supplemental information: Expanded view and Appendix (see: <https://www.embopress.org/page/journal/17574684/authorguide#expandedview>)

In this case, I would suggest the following:

- the 4 supplemental figures should be changed to EV figures and labelled and called out in the text as such: Fig. EV1 etc. Legends must be provided in the main article and individual figure files provided as separate files (like now).
- the 3 supplemental tables should be relabelled Dataset EV1 and so on, the call outs updated accordingly in the main text and in the datasets themselves
- the supplementary materials and methods should be moved into the main article file and the references added to the main reference list

3) Figures:

- in figure 2U, the western blot bands are too tightly cropped round the signal. In addition, please indicate MW for all western blots.
- figure 6D, please check the joining lines between main image and zoomed in, I think there's a mistake.
- make sure that all figures are provided in high resolution and as single page A4 format portrait orientation

4) Source Data:

We encourage the publication of source data, particularly for electrophoretic gels, blots, but also microscopy images with the aim of making primary data more accessible and transparent to the reader. Would you be willing to provide a PDF file per figure that contains the original, uncropped and unprocessed scans of all or key gels used in the figure (including molecular weight markers)? The PDF files should be labeled with the appropriate figure/panel number (1 file/figure), and should have molecular weight markers; further annotation may be useful but is not essential. It would also be useful to get the data behind the Kaplan-Meier survival graph.

The PDF files will be published online with the article as supplementary "Source Data" files. If you have any questions regarding this just contact me.

5) In the main manuscript file, please do the following:

- correct/answer the track changes suggested by our data editors by working from the uploaded

document (please keep the changes visible in track mode)

- limit keywords to 5 maximum
- change "Experimental procedures" to "Materials and Methods"
- in M&M, provide the antibody dilutions that were used for each antibody in all experiments
- in M&M, the PCR and Western Blots experimental details are not sufficient
- delete the running title
- in M&M, the statistical paragraph should reflect all information that you have filled in the Authors checklist, especially regarding randomisation, blinding, replication.
- indicate in legends exact $n=$ and exact $p=$ values, not a range, along with the statistical test used. Some people found that to keep the figures clear, providing an Appendix table Sx or an EV Table with all exact p -values was preferable. You are welcome to do this if you want to.
- in M&M, include a statement that informed consent was obtained from all human subjects and that the experiments conformed to the principles set out in the WMA Declaration of Helsinki and the Department of Health and Human Services Belmont Report. This is in the author checklist but missing from the paper.
- remove "data not shown" p6, 12, 13 and 14. As per our guidelines, on "Unpublished Data" the journal does not permit citation of "Data not shown". All data referred to in the paper should be displayed in the main or Expanded View figures. "Unpublished observations" may be referred to in exceptional cases, where these are data peripheral to the major message of the paper and are intended to form part of a future or separate study, the names of the persons that reported the observation should be listed in brackets. Personal communications (Author name(s), personal communications) must be authorised in writing by those involved, and the authorisation sent to the editorial office at time of submission.
- Similarity check: we noticed an overall 8% of similarity between your paper and a previously published article authored by you in Cell Reports in 2017. Please change phrases and sentences to avoid issues of self-plagiarism, at least in the main text of the article (It is more accepted in the M&M section)

6) Funding:

Please make sure to indicate in our submission system all sources of funding including grant numbers and to whom they are allocated.

7) For more information: There is space at the end of each article to list relevant web links for further consultation by our readers. Could you identify some relevant ones and provide such information as well? Some examples are patient associations, relevant databases, OMIM/proteins/genes links, author's websites, etc...

8) The Paper Explained: EMBO Molecular Medicine articles are accompanied by a summary of the articles to emphasize the major findings in the paper and their medical implications for the non-specialist reader. Please provide a draft summary of your article highlighting

- the medical issue you are addressing = Problem
- the results obtained = Results
- their clinical impact = Impact

9) A Conflict of Interest statement should be provided in the main text under the header: Conflict of Interest

10) Every published paper now includes a 'Synopsis' to further enhance discoverability. Synopses are displayed on the journal webpage and are freely accessible to all readers. They include a short stand first (maximum of 300 characters, including space) as well as 2-5 one sentence bullet points that summarise the paper. Please write the bullet points to summarise the key NEW findings. They should be designed to be complementary to the abstract - i.e. not repeat the same text. We encourage inclusion of key acronyms and quantitative information (maximum of 30 words / bullet point). Please use the passive voice. Please attach these in a separate file or send them by email, we will incorporate them accordingly.

You are also encouraged to suggest a striking image or visual abstract to illustrate your article. If you do please provide a jpeg file 550 px-wide x (250-400)-px high.

11) As part of the EMBO Publications transparent editorial process initiative (see our Editorial at <http://embomolmed.embopress.org/content/2/9/329>), EMBO Molecular Medicine will publish online a Review Process File (RPF) to accompany accepted manuscripts.

In the event of acceptance, this file will be published in conjunction with your paper and will include the anonymous referee reports, your point-by-point response and all pertinent correspondence relating to the manuscript. Let us know whether you agree with the publication of the RPF.

12) Please note that we now mandate that all corresponding authors list an ORCID digital identifier. This takes less than 90 seconds to complete. We encourage all authors to supply an ORCID identifier, which will be linked to their name for unambiguous name identification.

13) Data and software availability:

To list the primary data generated in your study, we would kindly ask you to include a formal "Data (and software) availability" section (after Materials & Methods) that follows the example below:

- [data type]: [full name of the resource] [accession number/identifier] ([doi or URL or identifiers.org/DATABASE:ACCESSION])

Ex: RNA-Seq data: Gene Expression Omnibus GSE1234
(<https://www.ncbi.nlm.nih.gov/geo/query/acc.cgi?acc=GSE1234>)

Ex. 2: Protein interaction AP-MS data: PRIDE PXD000123
(<http://www.ebi.ac.uk/pride/archive/projects/PXD000123>)

Please submit your revised manuscript within two weeks.

I look forward to receiving a new revised version of your manuscript as soon as possible.

Yours sincerely,

Celine Carret

Celine Carret, PhD

Senior Editor
EMBO Molecular Medicine

*** Instructions to submit your revised manuscript ***

To submit your manuscript, please follow this link:

Link Not Available

- 1) a .docx formatted version of the manuscript text (including Figure legends and tables)
- 2) Separate figure files*
- 3) supplemental information as Expanded View and/or Appendix. Please carefully check the authors guidelines for formatting Expanded view and Appendix figures and tables at <https://www.embopress.org/page/journal/17574684/authorguide#expandedview>
- 4) a letter INCLUDING the reviewer's reports and your detailed responses to their comments (as Word file).
- 5) The paper explained: EMBO Molecular Medicine articles are accompanied by a summary of the articles to emphasize the major findings in the paper and their medical implications for the non-specialist reader. Please provide a draft summary of your article highlighting
 - the medical issue you are addressing,
 - the results obtained and
 - their clinical impact.This may be edited to ensure that readers understand the significance and context of the research. Please refer to any of our published articles for an example.
- 6) For more information: There is space at the end of each article to list relevant web links for further consultation by our readers. Could you identify some relevant ones and provide such information as well? Some examples are patient associations, relevant databases, OMIM/proteins/genes links, author's websites, etc...
- 7) Every published paper now includes a 'Synopsis' to further enhance discoverability. Synopses are

displayed on the journal webpage and are freely accessible to all readers. They include a short stand first (maximum of 300 characters, including space) as well as 2-5 one sentence bullet points that summarise the paper. Please write the bullet points to summarise the key NEW findings. They should be designed to be complementary to the abstract - i.e. not repeat the same text. We encourage inclusion of key acronyms and quantitative information (maximum of 30 words / bullet point). Please use the passive voice. Please attach these in a separate file or send them by email, we will incorporate them accordingly.

You are also welcome to suggest a striking image or visual abstract to illustrate your article. If you do please provide a jpeg file 550 px-wide x 400-px high.

8) A Conflict of Interest statement should be provided in the main text

9) Please note that we now mandate that all corresponding authors list an ORCID digital identifier. This takes <90 seconds to complete. We encourage all authors to supply an ORCID identifier, which will be linked to their name for unambiguous name identification.

Currently, our records indicate that there is no ORCID associated with your account.

Please click the link below to provide an ORCID:

Link Not Available

10) The system will prompt you to fill in your funding and payment information. This will allow Wiley to send you a quote for the article processing charge (APC) in case of acceptance. This quote takes into account any reduction or fee waivers that you may be eligible for. Authors do not need to pay any fees before their manuscript is accepted and transferred to our publisher.

Photos 400-800 DPI

*Additional important information regarding figures and illustrations can be found at <http://bit.ly/EMBOPressFigurePreparationGuideline>

The system will prompt you to fill in your funding and payment information. This will allow Wiley to send you a quote for the article processing charge (APC) in case of acceptance. This quote takes into account any reduction or fee waivers that you may be eligible for. Authors do not need to pay any fees before their manuscript is accepted and transferred to our publisher.

***** Reviewer's comments *****

Referee #1 (Remarks for Author):

I would like to thank the authors for providing a robust and thorough response to my previous comments. They have addressed all of my concerns, and I believe that the manuscript now contains strong evidence for the role of glufosinate in altering the phenotype of tumoral macrophages from M1 to M2-like state.

Referee #2 (Comments on Novelty/Model System for Author):

No concerns with the model systems used. They have multiple immunocompetent models and show consistent data throughout.

Referee #2 (Remarks for Author):

The authors have addressed my concerns adequately. In fact the paper is quite interesting. I was surprised that macrophage depletion with anti-CSF1R did not suppress metastases (Fig 5E) but upon a literature review there are multiple other reports with the same finding in the LLC model. Further this macrophage depletion study supports the overall conclusion of the paper. I have no concerns about the manuscript and think it will be received well by the community.

1st Aug 2020

Dear Prof. Castegna,

Thank you for working with us on the revised files. We are pleased to inform you that your manuscript is accepted for publication and is now being sent to our publisher to be included in the next available issue of EMBO Molecular Medicine.

Please read below for additional IMPORTANT information regarding your article, its publication and the production process.

Congratulations on your interesting work,

Celine Carret

Celine Carret, PhD
Senior Editor
EMBO Molecular Medicine

Follow us on Twitter @EmboMolMed
Sign up for eTOCs at embopress.org/alertsfeeds

*** ** IMPORTANT INFORMATION ** **

SPEED OF PUBLICATION

The journal aims for rapid publication of papers, using the advance online publication "Early View" to expedite the process: A properly copy-edited and formatted version will be published as "Early View" after the proofs have been corrected. Please help the Editors and publisher avoid delays by providing e-mail address(es), telephone and fax numbers at which author(s) can be contacted.

Should you be planning a Press Release on your article, please get in contact with embomolmed@wiley.com as early as possible, in order to coordinate publication and release dates.

LICENSE AND PAYMENT:

All articles published in EMBO Molecular Medicine are fully open access: immediately and freely available to read, download and share.

EMBO Molecular Medicine charges an article processing charge (APC) to cover the publication costs. You, as the corresponding author for this manuscript, should have already received a quote with the article processing fee separately. Please let us know in case this quote has not been received.

Once your article is at Wiley for editorial production you will receive an email from Wiley's Author Services system, which will ask you to log in and will present you with the publication license form

for completion. Within the same system the publication fee can be paid by credit card, an invoice, pro forma invoice or purchase order can be requested.

Payment of the publication charge and the signed Open Access Agreement form must be received before the article can be published online.

PROOFS

You will receive the proofs by e-mail approximately 2 weeks after all relevant files have been sent to our Production Office. Please return them within 48 hours and if there should be any problems, please contact the production office at embopressproduction@wiley.com.

Please inform us if there is likely to be any difficulty in reaching you at the above address at that time. Failure to meet our deadlines may result in a delay of publication.

All further communications concerning your paper proofs should quote reference number EMM-2019-11210-V3 and be directed to the production office at embopressproduction@wiley.com.

Thank you,

Celine Carret, PhD
Senior Editor
EMBO Molecular Medicine

Corresponding Author Name: Massimiliano Mazzone

Journal Submitted to: EMBO Molecular MEDICINE

Manuscript Number: EMM-2019-11210